# Mesenchymal Stem Cell-Derived Exosomes in Ophthalmology: A Comprehensive Review

**DOI:** 10.3390/pharmaceutics15041167

**Published:** 2023-04-06

**Authors:** Kevin Y. Wu, Hamza Ahmad, Grace Lin, Marjorie Carbonneau, Simon D. Tran

**Affiliations:** 1Department of Surgery—Division of Ophthalmology, University of Sherbrooke, Sherbrooke, QC J1G 2E8, Canada; 2Faculty of Medicine, McGill University, Montreal, QC H3A 0G4, Canada; 3Faculty of Medicine, University of Montreal, Montreal, QC H3T 1J4, Canada; 4Faculty of Dental Medicine and Oral Health Sciences, McGill University, Montreal, QC H3A 1G1, Canada

**Keywords:** ophthalmology, ocular pharmacology, anterior segment diseases, posterior segment diseases, cell-based drug delivery systems, MSCs-based cell therapy, MSC-derived exosome, exosomes-based drug delivery, tissue repair and regeneration

## Abstract

Over the past decade, the field of mesenchymal stem cell (MSC) therapy has exhibited rapid growth. Due to their regenerative, reparatory, and immunomodulatory capacities, MSCs have been widely investigated as therapeutic agents in the cell-based treatment of chronic ophthalmic pathologies. However, the applicability of MSC-based therapy is limited by suboptimal biocompatibility, penetration, and delivery to the target ocular tissues. An emerging body of research has elucidated the role of exosomes in the biological functions of MSCs, and that MSC-derived extracellular vesicles (EVs) possess anti-inflammatory, anti-apoptotic, tissue repairing, neuroprotective, and immunomodulatory properties similar to MSCs. The recent advances in MSCs-derived exosomes can serve as solutions to the challenges faced by MSCs-therapy. Due to their nano-dimensions, MSC-derived exosomes can rapidly penetrate biological barriers and reach immune-privileged organs, allowing for efficient delivery of therapeutic factors such as trophic and immunomodulatory agents to ocular tissues that are typically challenging to target by conventional therapy and MSCs transplantation. In addition, the use of EVs minimizes the risks associated with mesenchymal stem cell transplantation. In this literature review, we focus on the studies published between 2017 and 2022, highlighting the characteristics of EVs derived from MSCs and their biological functions in treating anterior and posterior segment ocular diseases. Additionally, we discuss the potential use of EVs in clinical settings. Rapid advancements in regenerative medicine and exosome-based drug delivery, in conjunction with an increased understanding of ocular pathology and pharmacology, hold great promise for the treatment of ocular diseases. The potential of exosome-based therapies is exciting and can revolutionize the way we approach these ocular conditions.

## 1. Introduction

In recent years, the field of mesenchymal stem cell (MSC) therapy has garnered widespread attention for its potential therapeutic utility in the treatment of ophthalmic diseases. Despite significant regenerative, reparatory, and immunomodulatory properties, MSC-based therapy faces several limitations. More specifically, suboptimal biocompatibility, penetration, and delivery to the target ocular tissues narrow the applicability of MSC-based therapies. To circumvent these challenges, researchers have shifted their focus to a new aspect of MSCs-their exosomes.

Exosomes are nano-sized vesicles that possess anti-inflammatory, anti-apoptotic, tissue-repairing, neuroprotective, and immunomodulatory properties, similar to their parent MSCs. The potential benefits of using MSC-derived exosomes as a drug-delivery system can be fully realized, as they may be able to better penetrate barriers such as the blood-retinal barrier (Figure 1), based on the extrapolation of their ability to penetrate the blood-brain barrier. Additionally, their cargo is protected from degradation, resulting in increased bioavailability in ocular tissue [1,2,3].

While the potential of MSC transplantation in the field of regenerative medicine is undoubtedly great, recent research suggests that treatment utilizing MSC-derived exosomes could provide several benefits over traditional MSC-based therapies. The alternative use of exosomes allows practitioners to circumvent potential risks associated with MSC-centered therapies. Such risks include allogeneic immunological rejection, unwanted differentiation, and obstruction of small vessels through intravenous MSC injection, and their avoidance is critical for optimizing treatment outcomes [4].

Here, we focus on recent advances in the study of MSC-derived exosomes, published between 2017 and 2023. We highlight the characteristics and biological functions of MSC-exosomes and explore their potential in the treatment of various ocular diseases of the anterior and posterior segments of the eye. Additionally, we examine the possibilities of exosome-based therapies in a clinical context and the challenges that require remediation in preclinical studies (in vitro and animal-based) to facilitate a smooth transition to clinical trials.

## 2. Overview of MSC-Derived Exosomes

Mesenchymal stem cell-derived exosomes (MSC-exosomes) have garnered considerable interest as potential novel therapeutic agents in the treatment of various traumatic, inflammatory, vascular, and degenerative ocular diseases. More specifically, a growing body of research has delineated the therapeutic utility of MSC-exosomes in the treatment of autoimmune uveitis, glaucoma, retinal injury, diabetic retinopathy, and optic neuropathy, among others [5,6,7]. While further studies are required, particularly in humans, MSC-exosome-centered therapies represent a promising new avenue in the management of refractory diseases of the eye.

### 2.1. Exosomes: Characteristics and Biogenesis

Extracellular vesicles are membrane-delimited particles that are important in the maintenance of cellular homeostasis and have been implicated in various pathologies. They can be subdivided into exosomes, microvesicles, and apoptotic bodies based on size and biosynthetic pathway. Broadly speaking, exosomes are a nano-sized subset of extracellular vesicles that range from 30 to 150 nm, consisting of a cargo enclosed by a lipid bilayer [8]. The exosomal cargo, which is representative of the cell of origin, consists of a heterogeneous assemblage of proteins, amino acids, metabolites, lipids, and nucleic acids (including, non-exhaustively, DNA, miRNA, mRNA, and lncRNA) [5].

Exosome biogenesis is initiated by the invagination and budding of the endosomal limiting membrane, forming multivesicular bodies (MVBs) that house intraluminal vesicles [9]. The exosomal cargo is sorted via the action of the endosomal sorting complex required for transport (ESCRT) and ESCRT-independent pathways [9,10]. Mediated by the interaction of Rab GTPases and SNARE proteins, the fusion of MVBs with the plasma membrane permits the secretion of cup-shaped exosomes into the extracellular environment [11,12]. The lipid bilayer protects the internal cargo from enzymatic degradation, maintaining biological potency and integrity, allowing the exosomes to persist in the ocular structure long after release [13]. Exosomes are thus optimized for long-distance transport in biological fluids, acting as mediators of intercellular communication upon internalization by target cells and the release of their protected cargo [13]. Modes of internalization include direct fusion with the target membrane, receptor-mediated endocytosis, macropinocytosis, and phagocytosis [14]. Interaction with the target cell and the liberation of the enclosed cargo into the intracellular environment produce changes in gene expression and cellular function [15]. While exosomes are produced by all cell types, MSCs have a far greater capacity for exosome production and secretion than cells derived from mesodermal lineages [16]. Adipose tissue, umbilical cord, bone marrow, and corneal stroma are the primary sources of MSC-exosomes to be used in the treatment of ocular diseases [17,18]. Their natural occurrence in bodily fluids and parent-acquired lipid bilayer renders them highly biocompatible [19,20]. As such, a growing body of research has delineated their potential as an effective drug delivery system and promising therapeutic agent for a range of refractory ocular diseases.

Figure 2 depicts the characteristics and biogenesis of exosomes.

### 2.2. Role of Exosomes in Cellular Communication

#### 2.2.1. The Transfer of Biomolecules by Exosomes and Its Role in Intercellular Communication

The unique composition of the enclosed cargo allows MSC-exosomes to elicit various responses in a range of target cell types. Increasing evidence demonstrates that miRNAs are important mediators of intercellular communication, with over 4000 distinct types detected in exosomal cargo [21,22]. The unique complement of miRNAs in a given MSC-exosome is reflective of the identity and state of the donor cell, with adipose, umbilical cord, and bone-marrow-derived MSC-exosomes housing a unique complement of miRNA types [19,20]. Upon release in recipient cells, miRNAs post-transcriptionally regulate gene expression by base-pairing with the 3′UTR of mRNA [23].

#### 2.2.2. The Immunomodulatory Potential of MSC-Exosomes in Immune-Mediated Ocular Diseases

The established and potential therapeutic value of MSC-exosomes is immense, as they possess immunomodulatory, immunosuppressive, pro-regenerative, pro-angiogenic, and anti-inflammatory properties. Although the eye is an immune-privileged site compared to other organs, immune-mediated diseases affecting the anterior and posterior segments can cause significant damage to ocular tissue [23]. MSC-exosomes have shown efficacy in treating various immune-mediated ocular disorders, such as Sjögren’s syndrome dry eye, corneal allograft rejection, and autoimmune uveitis, by modulating the overactive immune response that characterizes these pathologies. Notably, the underlying mechanism is similar in all these diseases and will be further detailed in the upcoming section (Section 3) [5]. More specifically, MSC-exosomes have been shown to promote the differentiation of M2 macrophages and regulatory T cells (Tregs) and to reduce T-lymphocyte and natural killer cell division [24].

### 2.3. Advantages of MSC-Exosomes in Ophthalmology

Therapies involving MSC-exosomes circumvent many of the risks associated with MSC transplantation, many of which hail from undesired cell differentiation, as well as the inflammatory response associated with cell-based therapies [5]. Downstream complications of such differentiation and inflammation can be severe and irreversible, leading to partial and complete vision loss [25]. Other risks associated with MSC transplantation include vitreous opacification, vitreous hemorrhage, proliferative vitreous retinopathy resulting in retinal detachment, allogeneic immunological rejection, retinal artery and vein occlusion, and malignant transformation [5,25]. These potential complications associated with MSC transplantation can be significantly reduced via the use of MSC-exosomes. Since the therapeutic benefits of MSC transplantation stem primarily from the secretion of soluble paracrine factors rather than from direct cell replacement, the use of MSC-exosomes provides a means of achieving similar efficacy with a more favorable safety profile [26].

In comparison to MSC transplantation, MSC-exosomes possess other favorable characteristics (Figure 3): [27].

MSC-exosomes can selectively act on specific tissues and cells due to their ability to express different types of surface molecules [28]. They can interact with and transfer their cargo to recipient cells, producing desirable changes in gene expression and cellular function [15].The protection afforded by the lipid bilayer means that MSC-exosomes can persist in the ocular structure for a long time. Specifically, the bilayer provides stability and structural rigidity and protects the enclosed cargo from premature enzymatic degradation [13].They are highly biocompatible due to their bilipid membrane acquired from the parent cells and their natural presence in the body fluids [19,20]. For instance, in comparison to MSC therapy, exosome-based therapies have a lower risk of teratoma formation, embolization, and graft versus host rejection [29,30].There is a possibility that they might be able to penetrate through biological barriers of the eye (i.e., blood-retinal and blood-aqueous barriers, tear film, corneal stromal, and vitreous) due to their small size and their bilipid membrane. (Figure 1) However, it is important to note that currently, no articles specifically address the ability of exosomes to penetrate the tear film, corneal, or other ocular barriers. Nevertheless, some research implies that exosomes can successfully traverse the blood-brain barrier (BBB) [31,32,33]. Furthermore, studies in other parts of the body have demonstrated exosomes’ capacity to overcome challenging barriers [34]. Based on this evidence, one might hypothesize that exosomes hold the potential to serve as a delivery platform for penetrating ocular barriers such as the tear film and corneal barriers. However, additional research is necessary to verify this hypothesis. This would offer greater versatility in terms of routes of administration and the ability to deliver a larger quantity of bioactive molecules to the target site.

As outlined above, the efficacy and safety profiles of MSC-exosomes have rendered them a promising novel therapeutic agent in the management of a variety of ocular diseases. While further study is undoubtedly required, they show considerable promise, especially in the treatment of ocular diseases that are refractory to current treatments.

MSC-exosomes can be biologically enhanced via the controlled manipulation of chemical, biological, and mechanical factors during parental cell development (MSCs) [14]. Additionally, MSC-exosomes can be optimized for use as a novel drug delivery vehicle, and their cargo can be genetically modified. Such optimization techniques are all active areas of research [5].

### 2.4. MSC-Exosome Isolation and Preservation

Isolation of MSC-exosomes is most commonly achieved via a differential or density-gradient ultracentrifugation [35]. While generally the most cost-effective of the various available methods, the procedure itself is time-consuming, labor-intensive, and may yield impurities [34]. Ultrafiltration, size exclusion chromatography, precipitation, and immune affinity capture are other isolation methods that may be used in place of ultracentrifugation [34]. Additionally, size-dependent (e.g., ExoChip) and immunoaffinity-based microfluidic technologies are rapidly gaining traction [36]. Although limited by the relative paucity of research compared to more traditional methods, they can achieve rapid separation and high purity and have relatively small sample requirements [34]. Following isolation, exosomes can be stored long-term at −80 °C, although concerns regarding changes in morphology and bioactivity have been reported [37]. Cryopreservation using liquid nitrogen and cryoprotective agents may circumvent these issues, thereby achieving superior preservation of exosome morphology and function [19].

### 2.5. Route of Administration-MSC-Exosome in Ophthalmology

MSC-exosomes are commonly administered via direct injection into the vitreous humour of the eye in animal models. Intravitreal injection is advantageous in that it maximizes intraocular levels of the therapeutic agent and is generally well-tolerated by the recipient [38]. Recent research has also demonstrated the efficacy of subconjunctival and periocular injection as an alternative administration route for MSC-exosomes. Topical application to the ocular surface, usually as eye drops, provides a minimally invasive mode of delivery, albeit one that requires substantially higher dosages due to rapid tear turnover and the protective epithelial barrier [39].

### 2.6. Bioengineering MSC-Exosomes for Enhanced Drug Delivery

The protective lipid envelope and small size of MSC-exosomes render them an effective vehicle for drug delivery. While natural MSC-exosomes containing endogenous biomolecules of interest can be utilized, bioengineered exosomes have broader therapeutic applications [40]. Transfection, electroporation, and overexpression are commonly used to load RNA, hydrophilic biological molecules, and proteins into MSC-exosomes [41]. Alternatively, transfection can be used to load drugs directly into MSCs; the secreted MSC-exosomes containing the drug are later isolated [15]. Surface protein modification can help target MSC-exosomes to specific recipient cells, improving treatment efficacy while limiting adverse systemic effects [41]. More specifically, the modification of targeting peptides on the exosomal surface via covalent modification, non-covalent modification, and genetic engineering improves the specificity of exosomes to their targets [42]. Targeting can also be achieved using iron oxide nanoparticles in conjunction with an external magnetic field to localize exosomes to a site of interest [42].

## 3. The Use of MSC-Derived Exosomes in Anterior Segment Diseases

### 3.1. MSC-Derived Exosomes for Corneal Regeneration

Corneal regeneration is a complex and dynamic process that normally involves inflammation, cellular proliferation, and extracellular matrix (ECM) remodeling. Following corneal damage, damaged or dead cells secrete cytokines and mediators that attract immune cells to the area and induce surviving keratocytes to proliferate and transform into fibroblasts, which migrate toward the region of injury. These fibroblasts are capable of secreting ECM and enzymes involved in ECM remodeling, such as matrix metalloproteinases (MMPs) and collagenase, and transforming into myofibroblasts, which aid in wound closure and contraction [43]. Persistent epithelial defects induce prolonged or excessive activation of myofibroblasts, which causes inappropriate deposition of collagen fibers, scar tissue formation, and corneal opacification [44]. Excessive inflammation and pathological angiogenesis may also contribute to corneal scarring and opacity [45]. The preservation of corneal transparency is an essential component of successful corneal regeneration, and therapeutic management of corneal wounds must promote healing while controlling inflammation, neovascularization, and disorder of the collagen-rich ECM.

Exosomes have been a subject of interest in the treatment of corneal pathologies because they have previously been shown to promote tissue repair and suppress inflammation [46,47]. In recent studies, MSC-derived exosomes have been found to promote corneal epithelial cell proliferation and migration in vitro, thereby accelerating re-epithelialization [18,48,49]. These findings have been successfully translated to in vivo animal studies, where corneal wound healing is significantly improved in MSC-derived exosome-treated eyes [18,48,50,51]. The presence of miR-21 within the MSC-derived exosomes was suggested to contribute to the corneal wound healing effects of exosomes through its regulation of the PTEN/PI3K/Akt pathway, as miR-21 inhibition induced a partial elimination of wound healing effects [50]. Moreover, MSC-exosomes appear to reduce inflammation and apoptosis following corneal damage, as seen by downregulation in RNA levels of proinflammatory cytokines such as IL-1β, IL-8, TNF-α, and NF-κB, and of pro-apoptotic protein Cas-8 [48]. These MSC-exosomes may also have anti-angiogenic properties, as seen by downregulation in the expression of angiogenesis-associated matrix metalloproteinases MMP-2 and MMP-9 and pro-angiogenic factor VEGF [48,49,52]. Studies have also elucidated the therapeutic benefits of MSC-exosomes on corneal stromal stem cells (CSSCs). CSSCs are the progenitor cells to keratocytes located within the limbal stroma and possess the therapeutic ability to reorganize disordered ECM, with the potential to restore transparency [53,54]. Adipose-derived MSC-exosomes have been shown to promote the growth of CSSCs and inhibit their apoptosis in a dose-dependent manner, suggesting potential use in ECM remodeling to reduce corneal opacity [52].

Autophagy, a process implicating the self-degradation of cellular components to maintain homeostasis through the degradation and recycling of parts, has previously been shown to help corneal impairment [55,56]. In a study by Ma et al. (2022), MSC-exosomes were combined with an autophagy activator (AA), Rapamycin (50 nM), and their effect on corneal regeneration was evaluated [51]. The AA was shown to exert similar positive effects on migration and proliferation in corneal epithelial cells as human umbilical MSC-exosome treatment, and the combined effect of the exosome-AA treatment was superior to exosomes or AA alone. The percentage of apoptotic cells was the lowest in the mouse eyes treated with Exo-AA, whereas it was highest in the group treated with a combination of exosomes and an autophagy inhibitor (AI). Haze grade and expression of proinflammatory genes TNF-α, IL-1β, IL-6, and CXCL-2 were also reduced in the Exo-AA group. Autophagy activators could therefore be an interesting potential supplement to exosome-based therapies.

Several animal experiments have tested the use of bioengineered hydrogels as scaffolds for corneal repair, modified for the sustained release of MSC-exosomes. Implantation of thermosensitive chitosan-based hydrogels (CHI) of sustained-released induced pluripotent stem cell-derived MSC-exosomes (iPSC-MSC-exos) in a rat corneal anterior lamellar injury model yielded higher corneal transparency with a downregulation in the expression of collagen, compared to non-treated eyes. Upon further analysis of the overexpressed miRNA in iPSC-MSC-exos, miR-432-5p was found to downregulate collagen biosynthesis in CSSCs through modulation of the translocation-associated membrane protein 2 (TRAM-2) [57]. A thermosensitive hydrogel of modified hyaluronic acid with di(ethylene glycol) monomethyl ether methacrylate (DEGMA) was developed for the controlled release of exosomes rich in miRNA-24-3p from adipocyte-derived MSCs. In a study on an alkali burn model in rabbits, this hydrogel was observed to resist clearance from blinking while forming a clear and uniform layer on the ocular surface. Its application resulted in improved corneal epithelial defect healing, reduced corneal stromal fibrosis, and decreased macrophage activation [58], and miRNA 24-3p is said to promote corneal epithelial cell migration and corneal repair [58]. Current explorations in treatments combining biosynthetic hydrogels with MSC-exosome delivery could potentially promise an alternative to the conventional penetrating keratoplasty, which is associated with various complications such as corneal graft rejection, wound leak, and endophthalmitis.

### 3.2. MSC-Derived Exosomes for Dry Eye Disease (DED)

Dry Eye Disease (DED), also referred to as keratoconjunctivitis sicca, is a complex condition characterized by reduced tear quality or insufficient tear production. Symptoms include persistent discomfort, visual disturbances, and tear film instability, resulting in inflammation and damage to the ocular surface. DED can be attributed to a range of causes, including age, medical conditions such as Sjogren’s syndrome, medication side effects, environmental and lifestyle factors, and hormonal changes. Treatments for DED vary depending on disease severity and range from artificial tears and ointments for mild cases to topical corticosteroids, immunosuppressants, and autologous tear therapy for those that are more severe [59].

However, each of these treatments has its own limitations. For example, the efficacy of treatment with artificial tears hinges on patient compliance, which may decline over time. Long-term use of topical steroids can lead to side effects such as increased eye pressure and cataracts. Autologous tear therapy is both costly and requires multiple visits to a healthcare setting. Additionally, DED can affect ocular drug delivery by reducing the dwell time of topical drugs and increasing the risk of systemic absorption, as well as increasing the tear turnover rate, which can reduce the efficacy of topical medications. Despite the challenges posed by current treatments for DED, there is hope on the horizon. MSC-derived exosomes are emerging as a highly promising therapeutic option, offering the potential to address the root causes of DED effectively [60].

#### 3.2.1. GVHD-Associated DED

Previously, MSC-exosomes have been shown to have immunomodulatory effects in mice with chronic graft-versus-host disease (cGVHD), significantly suppressing Th17 expression while inducing Treg expression and prolonging survival of the mice, suggesting the potential to treat cGVHD and attenuate associated complications [61,62]. More recent studies on the topical application of MSC-exosomes in dry eye disease in mice have shown promising results for the future of cGVHD-associated DED management. Eyes treated with MSC-exosomes had increased tear secretion with longer tear break-up time, preservation of goblet cells, and fewer corneal defects [61,63,64,65]. Improvements were seen in the structure of the epithelium, with better morphological features of the microvilli and increased quantities of chondriosomes and desmosomes. Inflammation was reduced, observed by downregulation in proinflammatory genes such as IL-6, IL-1β, IL-17α, and Cd86, decreased levels of dendritic cells with suppression of MHC II expression; suppression of NLRP3 inflammasome activation; and a shift in the population of M1 proinflammatory macrophages to M2 anti-inflammatory macrophages [39,63,65,66]. The miRNA miR-204, important in ocular development and responsible for the suppression of the IL-6/IL-6R/Stat3 pathway, was identified as one of the most abundant miRNAs in MSC-exosomes, and subsequent miR-204 knockdown-induced reversal of the M1 to M2 transformation and abolition of therapeutic effects [39]. Furthermore, Ma et al. (2022) demonstrated that the addition of ascorbic acid to MSC-exosomes enhanced their therapeutic effects in DED by improving reactive oxygen species (ROS) scavenging [66].

A recent phase 1/2 single-arm clinical trial (NCT04213248) investigated the efficacy of treatment with MSC-derived exosomes in patients with GVHD-associated DED refractory to topical steroids, artificial tears, and autologous serum. Fourteen patients (28 eyes) received human umbilical MSC-derived exosomes administered as eye drops and dispensed at 10 µg/µL, to be administered four times a day per eye for two weeks. Results after 14 days revealed a significant reduction in corneal epithelial damage and improvement in epithelial recovery, with patient-reported relief of dry eye symptoms such as burning, stinging, redness, and crusting. No effects on intraocular pressure (IOP) were observed, nor were any complications related to the MSC-exosome treatment. Such findings suggest that the short-term use of MSC-exosomes may present a safe and effective treatment modality for severe GVHD-associated DED [39].

Similarly, a phase 1/2 single-arm clinical trial (NCT04213248), to be completed by May 2023, is investigating 27 subjects affected by dry eye symptoms from cGVHD. The group will receive artificial tears for two weeks to establish a baseline, followed by the umbilical MSC-derived exosome intervention dosed at 10 µg/drop, administered four times a day for 14 days. The progression of dry eye will be measured at a follow-up 12 weeks post-treatment. The study is currently in the process of recruiting participants [67].

#### 3.2.2. Sjogren’s Syndrome Dry Eye (SSDE)

Studies have been conducted on understanding the role and mechanisms of MSC-derived exosomes in the potential management of SSDE. Significant enrichment of miR-21 has been observed in MSC-exosomes, which could possibly play an important role in exosome-related immune regulation [68]. Exosomes derived from MSCs have been found to restore the balance in mi-RNA-125b-5p and miRNA-155-5p expression in CD4+ T cells, where miRNA-125b regulates B-cell differentiation by inhibiting PRDM1 translation and miRNA-155 is linked with cytokine production and CD8+ T cell proliferation [69,70,71]. Furthermore, MSC-derived exosomes in mice reduced signs of SSDE and promoted the repair, regeneration, and function of salivary and lacrimal glands [68]. Increased saliva flow rate and reduced tissue damage have also been observed in mice eyes treated with exosomes derived from olfactory ecto-MSC, which have been shown to secrete IL-6 [72]. IL-6 increases the immunosuppressive capacity of myeloid-derived suppressor cells by activating the STAT3 pathway [72].

### 3.3. MSC-Derived Exosomes for Corneal Clouding in Mucopolysaccharidosis

In patients with mucopolysaccharidosis IVA, a deficiency in the *N*-acetylgalactosamine-6-sulfate sulfatase (GALNS) enzyme results in an accumulation of GAGs, keratan sulfate, and chondroitin-6-sulfate in the lysosomes of all tissues. Such accumulation in the cornea causes corneal clouding and consequent visual impairment [73]. Research has illustrated the efficacy of MSC transplantation in reducing glycosaminoglycans (GAGs) accumulation and corneal haze in mucopolysaccharidosis (MPS) VII in the model. A recent study evaluated the potential of human umbilical MSC-derived EVs to transfer the GALNS to deficient cells to treat MPS IVA [74]. The study showed that the UMSC-EVs secrete active GALNS that is readily taken up by deficient cells in vitro [75]. However, the quantity of GALNS present in the EV isolates was low, requiring a line of UMSC to be transformed to constitutively express the GALNS enzyme [75]. The transfection technique has yet to be optimized for consistency and stability. These findings suggest promising new therapeutic avenues for the treatment of MPS in avascular tissues such as the cornea.

### 3.4. MSC-Derived Exosomes for Glaucoma

Glaucoma is a group of eye diseases that cause progressive vision loss and blindness via damage to the optic nerve. Globally, glaucoma is a leading cause of irreversible blindness. Currently, the primary treatment for glaucoma involves the management of intraocular pressure (IOP), but this avenue is incapable of addressing permanent damage to the retinal ganglion cells and the optic nerve that characterizes advanced stages of the disease. Additionally, vision loss can progress significantly in cases of normotensive glaucoma and controlled IOP. An ongoing investigation into neuroprotective strategies for glaucoma attempts to address these issues. From this, MSC-derived exosomes have emerged as a promising approach for promoting neuroprotection and delivering neuroprotective molecules to the posterior segment of the eye [76].

#### 3.4.1. MSC-Derived Exosomes for Glaucomatous Optic Neuropathy

Several studies have demonstrated the neuroprotective effects of MSC-derived exosomes on retinal ganglionic cells (RGCs). In the glaucomatous and ONC animal model, increased survival of treated RGCs compared to non-treated RGCs has been observed [29,30,77,78,79]. This differential survival has been attributed to several proposed mechanisms, including suppression of *cis* p-tau accumulation and subsequent RGC apoptosis [29], miRNA modulation, and secretion of neurotrophic factors (NTFs) [29,77,80]. Human placental MSC-EVs have been shown to attenuate hypoxic injury of immortalized R28 retinal progenitor cells in vitro through activation of the LONP1/p62 signaling pathway and repair of mitochondrial function [81]. In vivo, the EVs promoted the expression of antioxidants Prdx2 and Prdx 5 [81]. Mead et al. (2020) proposed TNF-α priming of MSCs to enhance the neuroprotective effects of the derived exosomes, with their study yielding significant improvement in neuroprotection and in quantities of exosomal NTFs such as PEDF, VEGF-A, and PDGF-AA [80]. They have previously demonstrated that the therapeutic benefits of MSC-exosomes halted 6 months after treatment and were completely absent between 9 and 12 months [29]. Their findings indicate that intravitreal injections would be required every few months to maintain the therapeutic window, which could potentially increase the risk of adverse effects and complications.

In a glaucomatous rat model, the neuroprotective effects of intravitreal administration of bone marrow MSC-exosomes were partially abolished when the exosomes were transfected with an Argonaute-2 (Ago2) inhibitor, which depletes the miRNA content [79]. The muted effect of exosomes following Ago2 knockdown suggests that the beneficial effects of MSC-exosomes are miRNA-dependent; proposed candidates include miR-21, miR-146a, and miR-17-92 [77]. The miRNA miR-17-92 has been found to be present within bone marrow MSC-exosomes and is involved in the downregulation of phosphatase and tensin homolog (PTEN) expression, which is an important suppressor of RGC axonal growth and survival [82,83]. The miRNA miR-146a targets the epidermal growth factor receptor, whose inhibition promotes RGC regeneration [84,85,86]. The miRNA miR-21 regulates PTEN expression and the EGFR pathway and has been shown to also affect astrocyte activation [30,87,88]. Interestingly, in a study by Li et al. (2018), inhibition of miR-21 promoted axonal regeneration through modulation of the EGFR/PI3K/AKT/mTOR pathway by preventing excessive astrocyte activation and glial scar progression [87]. The therapeutic role of miR-21, therefore, remains in question, and further studies would be needed to determine its importance in RGC regeneration. Bone marrow-derived MSC-exosomes have also been shown to promote neurogenesis with moderate RGC axonal regeneration significant at short distances from the site of the lesion (<1 mm), whereas umbilical MSC-exosomes have been shown to have no axogenic effect [30]. This discrepancy was suggested to be attributed to the different miRNA compositions of bone marrow MSC-exosomes and umbilical MSC-exosomes, implying that further studies in miRNA content could help us better understand the therapeutic effects of exosomes [30].

#### 3.4.2. MSC-Derived Exosomes for Intraocular Pressure (IOP) Lowering Effect

While RGC neuroprotection and neurogenesis have been studied extensively, an alternative potential therapeutic target of MSC-derived exosomes in glaucoma has been suggested. The trabecular meshwork cells (TMCs) in the iridocorneal angle drain the circulating aqueous humor in the anterior chamber [89]. Oxidative stress damage, mitochondrial dysfunction, and the accumulation of metabolites are all important contributors to TMC damage, which increases outflow resistance and leads to elevated IOP [90]. In a study by Li et al. (2021), the effects of exosomes derived from human bone marrow MSCs on in vitro human trabecular meshwork cells (hTMCs) under oxidative stress were investigated [91]. After exposure to H_2_O_2_ (0.1 mM), hTMCs pretreated with exosomes showed decreased production of intracellular reactive oxygen species (iROS) and downregulation of proinflammatory factors IL-1α, IL-1β, IL-6, and IL-8. There was also upregulation of matrix metalloproteinases MMP-2 and MMP-3, which regulate extracellular matrix remodeling with the potential to increase aqueous outflow capacity [92]. These findings suggest that MSC-exosome therapies may potentially alleviate trabecular meshwork dysfunction induced by oxidative stress, which could ultimately prevent the increase in intraocular pressure (IOP) and slow down the progression of glaucoma. Moreover, specific miRNAs have been linked to the mechanism of action. The miRNA miR-126-5p was downregulated in the treated group, implying a possibly reduced risk of glaucoma, as the miR-126 family of miRNA has previously been identified as upregulated in the tears of patients diagnosed with open-angle glaucoma [93]. In addition, miR-3529-3p was upregulated in the treated group, while its target gene, CXCL5, was downregulated. The inflammatory chemokine CXCL5 has previously been shown to be significantly elevated in the aqueous humor of patients with glaucoma [94].

Table 1 provides a summary of the potential of MSC-derived exosomes in treating anterior segment diseases and glaucoma.

## 4. The Use of MSC-Derived Exosomes in Posterior Segment Diseases and Uveitis

### 4.1. Retinitis Pigmentosa

Retinitis pigmentosa (RP) is a group of hereditary disorders that impact the retina, resulting in progressive vision loss. The loss is caused by the degeneration of the photoreceptor cells, primarily rods, due to various genetic mutations. For most patients suffering from retinitis pigmentosa (RP), there is no cure. Only a limited number of RP patients with the RPE65 gene mutation can receive the targeted gene therapy, known as voretigene neparvovec, and it has the potential to halt and cure the disease. This is because voretigene neparvovec is specifically designed to address the RPE65 mutation, which is only present in 0.3% to 1% of the patients suffering from RP, and its effectiveness relies on the presence of this particular genetic variant. Most RP patients must rely on conventional treatment methods, such as vitamin A supplementation, shielding from sunlight, visual aids, and medical or surgical interventions. Unfortunately, these conventional treatments often only provide limited benefits, primarily aimed at managing symptoms, preventing eye complications, and in some cases slowing the progression of the disease. They are unable to cure RP or restore a patient’s visual acuity to a satisfactory level. Given the broad and heterogeneous causes of RP, researchers have shifted their focus to preventing retinal degeneration by targeting the posterior eye. This can be achieved through the administration of neuroprotective agents with properties such as neurotrophic, anti-apoptotic, and antioxidant or through cell-based therapy, such as mesenchymal stem cell transplantation (MSCT). These treatments aim to prevent photoreceptor death, promote its survival, and potentially regenerate functional cells and tissues [96].

To understand the mechanisms underlying the successful prevention of photoreceptor apoptosis through mesenchymal stem cell transplantation (MSCT), Deng and colleagues (2020) used mouse bone marrow MSCT in an *N*-methyl-*N*-nitrosourea (MNU) driven photoreceptor injury mouse model [97]. Intravitreal MSC counteracted photoreceptor apoptosis and alleviated retinal morphological and functional degeneration as expected. However, the effects of MSCT were inhibited if mesenchymal stem cell exosome (MSC-exosomes) generation was blocked. Furthermore, isolated intravitreal injection of the MSC-exosomal miR-21 effectively suppressed MNU-provoked photoreceptor injury even after 1–2 months post-injection, highlighting a long-term therapeutic benefit of MSC-exosomes in treating photoreceptor apoptosis. Liu and colleagues (2019) conducted similar experiments to explore the possibility of using human bone marrow MSC-exosomes as an option to attenuate neuroinflammation and promote the survival of photoreceptors in RP. After treating Rd10 mutated mice with Msc-exosomes, the researchers concluded that MSC-exosomes protected the neuroretinal function, with improvements being observed on both electroretinogram (ERG) and optokinetic tracking response (OKT). Furthermore, MSC-exosomes suppressed the degeneration of photoreceptors with increased thickness of the retinal outer nuclear layer, decreased Tunel positive photoreceptor nuclei, and significantly inhibited expression of pro-inflammatory cytokines, indicating the relief of neuroinflammation. More recently, Zhang and colleagues (2022) investigated the use of MSCs-exosomes as a treatment option for RP and built upon the findings of the previous studies [98]. Intravitreal treatment of mice with MSC-exosomes increased the survival of photoreceptors, preserved their structure, and improved visual function. In addition, the findings of this study also showed that the treatment inhibited inflammation through overexpression of the miR-146a-Nr4a3axis, which was a target of the MSC-exosomes. All three studies elucidated the possible tremendous long-term benefits of using MSC-exosomes in the treatment of RP. However, these preclinical studies were conducted using animal models, and only future clinical studies will answer whether the same effects can be achieved in humans.

### 4.2. Diabetic Retinopathy

Diabetic retinopathy is a chronic eye condition that involves progressive damage to the blood vessels of the retina. Two types of diabetic retinopathy exist non-proliferative diabetic retinopathy (NPDR) and proliferative diabetic retinopathy (PDR). NPDR is characterized by elevated vascular permeability and blockage, leading to the formation of micro-aneurysms, dot and blot hemorrhages, cotton wool spots, and hard exudates. PDR, which occurs in the advanced stages of diabetic retinopathy, is caused by ongoing damage to the retinal blood vessels and results in significant retinal ischemia. The ischemic retinal tissue releases pro-angiogenic factors, including the vascular endothelial growth factor (VEGF), which stimulate the growth of new, abnormal blood vessels. Such abnormalities can give rise to vision-threatening complications, such as neovascularization of the disc and retina, causing vitreous hemorrhage and tractional retinal detachment, or neovascularization of the iris and angle, resulting in glaucoma. The primary focus of the management of PDR is inhibiting the activity of VEGF in ischemic tissue through laser photocoagulation or intravitreal anti-VEGF injections, which work by binding to VEGF and preventing it from interacting with its receptor [78].

Several preclinical studies have explored the therapeutic potential of MSC-exosomes to treat diabetic retinopathy (DR) and its associated complications. Safwat and colleagues (2018) investigated the potential use of MSC-exosomes in a streptozotocin-induced diabetes mellitus (DM) rabbit model [99]. MSC-exosomes isolated from the adipose tissue of rabbits were injected via different routes (intravenous (IV), subconjunctival (SC), and intraocular (IO)). Depending on the route of administration, different therapeutic results were observed. At 12 weeks post-administration, the IV route resulted in a retinal regeneration marked by an irregular ganglionic layer and increased retinal thickness. In contrast, following SC administration, retinal cellular components were organized in well-defined layers. Notably, IO administration regenerated the retina into well-defined layers that were morphologically and functionally analogous to those of a normal retina. In addition, the authors also explored the role of micRNA-222 in DR since micRNA-222 is involved in downregulating angiogenesis through the inhibition of signal transducer and activator of transcription 5A (STAT5A). The findings of this study revealed that micRNA-222, in fact, was under-expressed in hyperglycemic conditions of DR, and a reduced expression of micRNA-222 in retinal tissue of STZ-induced diabetes resulted in increased retinal damage and extensive hemorrhage in different layers of the retina. However, the retinal damage was ameliorated through MSC-exosomal transfer of micRNA-222 [79]. Another study conducted by Zhang and colleagues (2019) investigated the role of micRNA-126 (miR-126) transfer via human umbilical cord mesenchymal stem cell (MSC)-derived exosomes (hUCMSC-Exos) in regulating hyperglycemia-induced retinal inflammation [7]. Similar to micRNA-222, miR-126 has been implicated in the regulation of angiogenesis by mediating inflammation and vascular development through the downregulation of inflammatory cytokines such as interleukin-6 and tumor necrosis factor-alpha (TNF-α). Thus, exosomes overexpressing miR-126 were intravitreally injected into diabetic rats in vivo, while high glucose-affected human retinal endothelial cells (HRECs) were cocultured with hUCMSC-exosomes in vitro. Results indicated an increase in hyperglycemia-induced inflammatory cytokines in both diabetic rats and HRECs. However, the administration of hUCMSC-exosomes effectively reversed inflammation. Compared to control hUCMSC-exosomes, hUCMSC-exosomes overexpressing miR-126 more successfully suppressed inflammation in diabetic rats, highlighting the key role of miR-126 in attenuating DR.

To provide more specific therapeutic targets for MSC-exosomes, studies have begun exploring and implicating individual microRNAs involved in the amelioration of DR. Li and colleagues (2021) highlighted the regenerative capability of mice bone marrow mesenchymal stem cells (BMSCs)-induced exosomal microRNA-486-3p (miR-486-3p) in DR. Retinal Muller cells of mice were injected with streptozotocin (STZ) for 3-months to create a diabetic model [100]. Histological confirmation for typical pathological features of DR was made based on the upregulation of Toll-like receptor 4 (TLR4) and nuclear factor-kappa B (NF-κB). Exposure to BMSC-exosomes in vitro inhibited oxidative stress, inflammation, and apoptosis and promoted proliferation in Muller cells. Furthermore, upregulating miR-486-3p or down-regulating TLR4 inhibited oxidative stress, inflammation, and apoptosis and promoted the proliferation of Muller cells, implicating TLR4 as a target of miR-486-3p. Building upon the findings of the previous study, Li and colleagues (2021) performed further experiments using hUCMSC-exosomes to highlight the role of microRNA-17-3p in targeting the signal transducer and activator of transcription 1 (STAT1) on inflammatory reaction and antioxidant injury of DR mice model [101]. Upon experimentation, the authors observed a decrease in miR-17-3p and an increase in STAT1 in the retinal tissues of DR mice. Furthermore, an overexpression of miR-17-3p enhanced exosomes led to a decrease in STAT1 expression, while a depletion of miR-17-3p enhanced exosomes exerted an inverse effect, implicating STAT1 as a potential target of miR-17-3p. Overall, injection of MSC-exosomes overexpressed with miR-17-3p reduced the blood glucose and HbAlc, increased body weight and Hb content, decreased inflammatory factors and VEGF, alleviated oxidative injury, inhibited retinal cell apoptosis in DR mice through inhibition of STAT1.

More recently, Gu and colleagues (2022) explored the regulatory effect of bone marrow mesenchymal stem cell (BMSC) exosomal miR-146a on inflammation in DR mice [102]. Specifically, the investigators explored the use of BMSC exosomal miR-146a to treat microglial cells in DR mice. The findings of the study demonstrated a reduction in the levels of proliferating cell antigen and B-cell lymphoma-2 in microglia of DR mice upon exposure to BMSC exosomal miR-146a. Moreover, BMSC exosomal miR-146a reduced the levels of inflammatory cytokines such as TNF-α, IL-1β, and IL-6, suggesting that miR-146a can successfully alleviate inflammation in DR. Interestingly, overexpression of TLR4 reversed the effects of miR-146a on the proliferation, apoptosis, and inflammation of microglia, highlighting an inverse association between miR-146a and TLR4. Another study, conducted by Ebrahim and colleagues (2022), explored the Wnt/b-catenin signaling pathway in DR using rat bone marrow-derived mesenchymal stem cell exosomes (BMMSCs) [103]. The experimenters allocated rats with STZ-induced DR into six different treatment groups and evaluated each group for protein expression concerned with oxidative stress, inflammation, and angiogenesis. Findings of the study demonstrated that blockage of the wnt/b-catenin pathway via intravitreal administration of BMMSC-exosomes resulted in a significant decrease in retinal mRNA markers indicative of oxidative stress, inflammation, and angiogenesis and vascular leakage in DR compared to diabetic controls. These effects were achieved by targeting the miR-129–5 P and miR-34a exosomal microRNA.

Unlike the previous studies, Cao and colleagues (2021) demonstrated the involvement of long non-coding RNA (lncRNAs) small nucleolar RNA host gene (SNHG7) in the pathogenesis of DR [104]. Experimenters utilized human retinal microvascular endothelial cells (HRMECs) treated with high glucose (HG) to establish a DR cell model. Interestingly, LncRNA SNHG7 downregulated miR-34a-5p, and overexpression of SNHG7 inhibited hyperglycemia-induced endothelial–mesenchymal transition (EndMT) and tube formation of HRMECs. However, these benefits were reversed by overexpression of miR-34a-5p, and a subsequent knockdown of miR-34a-5p repressed HG-induced EndMT and tube formation. Overall, based on the findings of the study, the authors concluded that the MSC-exosomal lncRNA SNHG7 suppressed EndMT and tube formation in HRMECs via miR-34a-5p/XBP1 downregulation.

### 4.3. Age-Related Macular Degeneration

Age-related macular degeneration (AMD) is a prevalent eye disorder affecting individuals over the age of 50 and is a major cause of vision loss and blindness in the elderly. It affects the macula and leads to difficulties with activities such as reading and recognizing faces. AMD can be classified into three stages: early, intermediate, and late. In early-stage AMD, medium-sized drusen deposits are present without pigment changes or vision loss. As AMD progresses to the intermediate stage, individuals may experience large drusen deposits and/or pigment changes, with possible mild vision loss, although many remain asymptomatic. Late-stage AMD is further divided into two forms: dry and wet. Dry AMD, the more prevalent form, exhibits a gradual progression, while wet AMD, characterized by abnormal blood vessel growth beneath the macula, can cause rapid vision decline due to fluid and blood leakage. This is known as choroidal neovascularization (CNV). There are several targets for the treatment of AMD, including reducing inflammation and drusen formations, improving RPE cell survival, inhibiting angiogenesis, and treating choroidal neovascularization (CNV) in wet AMD. The treatment for AMD depends on its type and severity. Dry AMD can be managed through monitoring and the use of nutritional supplements, while wet AMD typically requires regular intravitreal injections of anti-VEGF drugs. Despite the widespread use of anti-VEGF treatments, not all patients respond favorably, and there are potential vision-threatening complications such as endophthalmitis and retinal detachment. The reliance on frequent intravitreal injections also puts a burden on patient compliance. MSC-derived exosomes offer potential advantages in addressing these issues: (1) they may significantly reduce the frequency of intravitreal injections due to better biocompatibility and longer duration of action resulting from protection against degradation (as depicted in Figure 3), and (2) they have the potential to be delivered topically instead of intravitreally due to their capacity to penetrate through barriers and target specific tissues (as depicted in Figure 3). Therefore, optimizing therapies that target both inflammation and neovascularization with the use of MSC-derived exosomes could provide a more effective and less burdensome treatment solution [101,102].

In 2013, Hajrasouliha and colleagues first demonstrated the benefits of exosomes in suppressing retinal vessel leakage and inhibiting choroidal neovascularization [105]. Even though the study utilized exosomes that were not derived from mesenchymal stem cells and a control group for specific molecules (proteins, lipids, mRNA, miRNA) was not established, the experimenters were able to elucidate the potential therapeutic effects of exosomes in age-related macular degeneration. In 2018, to study the effects of human umbilical cord MSC-exosomes on age-related macular degeneration (AMD) and the concomitant development of choroidal neovascularization (CNV), He and colleagues (2018) used blue light injured human retinal pigment epithelial (RPE) cell model alongside laser-induced choroidal neovascularization (CNV) rat model [106]. RPE cells exposed to blue light were cocultured with hUCMSCs in vitro while mice were injected with different doses of MSC-exosomes intravitreally to observe and compare their effects on CNV. In vitro, MSCs-derived exosomes downregulated the mRNA and protein expression of VEGF-A in RPE cells after blue light stimulation. In vivo, MSCs-exosomes reduced damage, distinctly downregulated VEGF-A, and gradually improved the histological structures of CNV for better visual function. Thus, the authors concluded that MSCs-exosomes ameliorated blue light stimulation in RPE cells and laser-induced retinal injury via downregulation of VEGF-A.

Another study conducted by Li and colleagues (2021) investigated the use of human umbilical cord-derived mesenchymal stem cells (hUCMSCs) in vivo and in vitro to attenuate subretinal fibrosis; a wound-healing response generated against choroidal neovascularization (CNV) in wet age-related degeneration (wet AMD) [101]. Laser-induced choroidal neovascularization (CNV) and subretinal fibrosis models were established in mice. Upon intravitreal injection of hUCMSC-exosome, alleviation in subretinal fibrosis was observed in vivo. In addition, hUCMSC-exosomes suppressed the migration of RPE cells and promoted the mesenchymal–epithelial transition via miR-27-3p. In addition, intravitreal injection of hUCMSC-exosome effectively ameliorated laser-induced CNV and subretinal fibrosis via the suppression of epithelial–mesenchymal transition (EMT) process. Both studies elucidated the use of MSC-exosomes in improving and reversing choroidal neovascularization secondary to age-related macular degeneration. In short, according to these preclinical studies, MSC-exosomes may offer an alternative treatment for wet AMD and CNV in the future, potentially reducing the need for frequent anti-VEGF injections currently prescribed by ophthalmologists.

### 4.4. Retinal Ischemia

Retinal ischemia is a pathological condition characterized by insufficient blood flow to the retina. This vascular insufficiency results in an inadequate supply of oxygen and nutrients, ultimately leading to retinal cell dysfunction and potential vision loss. Multiple factors can contribute to the development of retinal ischemia, including both systemic and ocular conditions. Systemic causes encompass diabetes mellitus, hypertension, and blood dyscrasias, such as sickle cell disease and leukemia. These conditions can lead to microvascular changes, impairing blood flow and increasing the risk of retinal ischemia. Ocular causes include retinal vascular occlusions, such as central or branch retinal artery occlusion (CRAO or BRAO), and central or branch retinal vein occlusion (CRVO or BRVO), as well as an ocular ischemic syndrome resulting from carotid artery stenosis. Inflammatory and degenerative eye diseases, such as uveitis and age-related macular degeneration, can also induce retinal ischemia by compromising ocular circulation. Treatment varies based on the underlying cause but often involves anti-VEGF medications to prevent abnormal blood vessel growth in the retina.

Several studies have investigated the use of MSC-exosomes in retinal ischemic injuries. A study conducted by Mosseiev et al. (2017) demonstrated the protective effect of human mesenchymal stem cells (hMSCs) administered intravitreally in the murine model [38]. Experimenters split 12 male C57BL/6 mice into three groups over a period of two weeks. Mice in groups 1 and 2 were placed in a closed chamber at 75% oxygen for five days to induce oxygen-induced retinopathy, followed by an administration of saline and MSC-exosomes, respectively. Mice in group 3 were kept in a standard room and injected with saline as a control. Based on the findings of the study, the authors concluded that hMSC-exosomes significantly reduced retinal thinning and neovascularization. In addition, the authors also suggested the possible role of paracrine factors and miRNAs as primary mediators of the observed therapeutic effect. In 2016, Yu and colleagues intravitreally injected mice with adipose-derived and human umbilical cord-derived MSC-exosomes to investigate the potential benefits of MSC-exosomes in a laser-induced injury and ischemia model [4]. Similar to the findings of the Mosseiev et al. (2017) study, MSC-exosomes reduced damage, inhibited apoptosis, and suppressed inflammatory responses to obtain a better visual function to nearly the same extent in vivo. In addition, downregulation of monocyte chemotactic protein (MCP)-1 in the retina was found after MSC-exosome injection, suggesting that MSC-exosomes ameliorate laser-induced retinal injury partially through down-regulation of MCP-1 (Yu et al., 2016). The studies on MSC-exosomes for retinal ischemic injuries demonstrate promising results and could potentially be extrapolated to other ischemic retinal diseases, such as retinopathy of prematurity, ocular ischemic syndrome, and retinal vein and artery occlusion.

After successfully highlighting the potential neuroprotective effects of an intravitreal injection of mesenchymal stem cells (MSC) and MSC-conditioned medium in retinal ischemia in rats, Mathew and colleagues (2019) concentrated on investigating and understanding the specific role of exosomes [107]. Using the R28 cell line derived from postnatal day 6 Sprague-Dawley rats, an in vitro oxygen-glucose deprivation (OGD) model of retinal ischemia was created. Exposure to Human MSCs (hMSCs) exosomes reduced cell death and attenuated loss of cell proliferation. Furthermore, enhanced functional recovery and decreased neuroinflammation and apoptosis were observed in the retinal ischemia rat model if MSC-exosome were injected into the vitreous humour 24 h post-ischemia development. Interestingly, MSC-exosomes were present in the vitreous humor for four weeks after intravitreal administration [107]. Another study conducted by Yu and colleagues (2022) explored the neuroprotective effects of human gingival MSCs (hGMSC) derived exosomes in retinal ischemia-reperfusion injury [108]. To study the effects, investigators injected hGMSC-exosomes into the vitreous of mice. To elucidate the potentiating effect of TNF-α, exosomes transfected with siRNA-maternally expressed gene 3 (siRNA-MEG3) were either stimulated by TNF-α (TG-exos) or unstimulated (G-exos). The results showed that IO of TG-exos into mice with retinal ischemia notably reduced inflammation and cell loss compared to G-exos IO injection. Similar results were observed in vitro. Additionally, it was found that miR-21-5p acted as a crucial factor in TG-exos for neuroprotection and anti-inflammation.

Finally, retinal ischemia can also occur in the case of retinal detachment. A study conducted by Ma and colleagues (2020) investigated the potential therapeutic effects of using rat bone marrow-derived MSC-exosomes in a rat retinal detachment model [109]. Sub-retinal administration of the MSC-Exosomes, performed at the time of retinal separation, highlighted that expression of proinflammatory cytokines at day seven was significantly reduced. Moreover, treatment with MSC-exosomes also suppressed photoreceptor cell apoptosis and maintained normal retinal structure when compared to control groups. The results of the study highlighted the potential therapeutic effects of MSC-exosomes on retinal ischemia secondary to retinal detachment.

### 4.5. Idiopathic Macular Hole

An idiopathic macular hole is a tear in the center of the retina. The formation of a macular hole is believed to be caused by pathological vitreoretinal traction. Metamorphopsia (distortion of the central vision), central visual loss, or central scotoma can be reported by patients with macular holes as common symptoms. The approach to treating this condition is based on its severity or stage. When the condition has progressed to stage 2 or above, characterized by a full-thickness defect from the internal limiting membrane to the retinal pigment epithelium, a surgery known as pars plana vitrectomy (PPV) is necessary. Unfortunately, PPV is an invasive surgical procedure that poses several risks, such as tearing or detachment of the retina, increased pressure within the eye, reinfection, and permanent vision loss. Furthermore, the postoperative period requires that the patient maintain a prone position, which can be uncomfortable and demands a great deal of compliance over the course of a week or more [110].

An alternative, less invasive treatment option is intravitreal ocriplasmin. Despite its less invasive nature, the procedure still involves certain risks associated with intravitreal injections and rare risks related to the ocriplasmin drug. These risks can include lens subluxation or phacodonesis, dyschromatopsia, transient changes in the electroretinogram, retinal tear or detachment, and decreased visual acuity [111].

Based on the therapeutic success of MSC-exosomes in treating several ocular pathologies, Zhang and colleagues (2018) explored the efficacy of using human umbilical cord-derived MSC-exosomes to promote the healing of large and refractory macular holes (MHs) [112]. Unlike previous preclinical studies utilizing animal models, this study recruited seven patients between the age of 51–71 years old with large idiopathic MHs. All patients underwent vitrectomy and an internal limiting membrane peeling. Following the surgery, two patients received MSCT therapy, whereas five patients received MSC-exosome treatment. Post-treatment, six MHs were closed, while one remained in a flat-open state. The best-corrected visual acuity (BCVA) was improved in five patients with MH closure and remained unchanged in one patient with MH closure who had a four-year history of MH. Overall, the findings of the study indicated the potential benefit of using MSC and MSC-exosome therapy to promote functional and anatomic recovery from MH. Furthermore, the authors observed that the MSC-exosome therapy was safer and easier to administer than MSCT therapy. This was partly because MSC therapy had a lower risk of developing proliferations and immune responses. Secondly, due to the small size of MSC and the various administrative routes (IV, IO, Intravitreal), MSC therapy did not require any additional surgeries. However, due to the lack of a control group and a limited number of patients, it could not be concluded if MSC-exosome therapy was superior to MSCT treatment for MH closure.

### 4.6. Uveitis

Uveitis is an ocular condition characterized by inflammation in the uvea, a pigmented layer of the eye that encompasses the iris, ciliary body, and choroid. The type of uveitis can be determined based on the part of the eye affected, with anterior, intermediate, posterior, and panuveitic forms being the most common types. The causes of uveitis can be diverse, with non-infectious idiopathic or autoimmune sources being the most prevalent, while infectious causes are typically less frequent but more severe. Most cases of non-infectious anterior uveitis are treated with glucocorticoid steroids, in the form of topical eye drops, sometimes in combination with oral therapy [113]. In certain cases, non-steroidal treatments, such as disease-modifying antirheumatic drugs and antimetabolite medications, may also be employed, depending on the specific cause and severity of the uveitis. While anterior uveitis can typically be effectively managed with topical medication, posterior uveitis poses a greater challenge due to its localized inflammation in the critical posterior segments of the eye, including the macula, optic nerve, and retinal vessels. If these important structures are affected, rapid vision loss and even blindness can result. To effectively treat posterior uveitis, more invasive administration methods, such as periocular or intravitreal injection, are often required, as topical eye drops cannot reach the posterior segment of the eye [114]. In these instances, the full potential of using MSC-derived exosomes as a drug-delivery system could be realized, given that they might effectively penetrate barriers, such as the blood-retinal barrier, based on the extrapolation from their ability to cross the blood-brain barrier [31,32,33]. Furthermore, their cargo would be protected from degradation, leading to increased bioavailability. This concept has been previously addressed in the article, as illustrated in Figure 3.

In 2014, Oh and colleagues first demonstrated the potential benefits of using mesenchymal stem cells to prevent experimental autoimmune uveitis (EAU). Intraperitoneal administration of hMSCs in mice demonstrated a reduction in the levels of proinflammatory cytokines in the eye. Analysis of draining lymph nodes (DLNs) indicated a marked suppression of Th1 and Th17 cells without a reduction in the levels of cytokines IL-1β, IL-6, IL-12, and IL-23 responsible for the differentiation of Th1/Th17 cells. In addition, hMSCs also increased the level of immunoregulatory cytokine IL-10 and IL-10-expressing B220+CD19+ cells. Together, these outcomes protected the retina from damage and attenuated experimental autoimmune uveoretinitis (EAU), suggesting hMSCs as a potential therapy.

However, since the therapeutic effects of MSCs are mediated via the transport and transfer of exosomes containing various miRNAs, investigators have begun focusing on exploring MSC-exosomes as potential therapeutic agents for EAU. A study conducted by Shigemoto-Kuroda and colleagues (2017) explored the use of MSC-exosomes for Type-1 Diabetes (T1D) and experimental autoimmune uveoretinitis (EAU) [115]. Exosomes were harvested from human bone marrow MSCs and administered intravenously into the tail vein of established mouse models for T1D and EAU. The findings of the study revealed that MSC-exosomes prevented the onset of T1D and EAU. Moreover, MSC-exosomes inhibited the activation of antigen-presenting cells and suppressed the activation of Th1 and Th17 cells. Overall, MSC-exosomes could serve as a potential treatment for T1D and EAU, with preclinical studies indicating they may be able to reduce the need for conventional treatments such as topical steroids and systemic immunosuppressants [115]. Similar findings were reported by Bai and colleagues (2017) when exploring the potential therapeutic effects of human umbilical cord-derived MSC-exosomes on EAU [116]. Using interphotoreceptor retinol-binding protein 1177–1191 peptide immunization, EAU was induced in rats. Post immunization, clinical signs of uveitis were used as a confirmation of EAU, and MSC-exosomes were administered in both eyes via periocular injections containing different doses of MSC-exosomes for seven consecutive days. In vivo administration of MSC-exosomes demonstrated a milder development of EAU compared to the control rats. Whereas in vitro chemotactic and lymphocytic proliferation assays demonstrated a marked reduction in the intensity of EAU following MSC-exosomes administration. Moreover, MSC-exosomes effectively inhibited the migration of CD4+T cells, neutrophils, NK cells, and macrophage cells and reduced the percentage of CD4+IFN-γ+ and CD4+IL-17+ cells in the retina, reducing unchecked inflammation. Interestingly, even though MSC-exosomes treatment reduced the concentration of T regulatory cells (Tregs), the reduction was not significant enough to eliminate the impending beneficial role of Tregs in the suppression of EAU.

In 2018, Xie and colleagues studied the effects of MSC-exosomes on alleviating EAU clinically and pathologically in a rat model. The experimenters randomly divided 12 rats into an experimental and control group, with each group receiving a periocular injection of MSC-exosomes and phosphate buffer, respectively [117]. Following MSC-exosome administration, decreased expression of CD68 cells was observed in the experimental rats compared to the control group. Even 15 days after administration of MSC-exosomes, retinal pathological scores were observed to remain significantly lower in the experimental group in comparison to the control. In addition, the number of Thl, Th17, and Tregs was also significantly decreased in the experimental group. When evaluating the retinal function using ERG, the experimental group performed significantly better than the control group 15 days post MSC-exosomes administration. Thus, the findings of this study further elucidated the role of MSC-exosomes in reducing the clinical and pathological manifestations of EAU, protecting retinal function, and down-regulating the proportion and infiltration of inflammatory cells in the eye [117]. More recently, Yongtao and colleagues (2021) studied the effects of MSC-exosomes on EAU through intravenous injection of hUCMSCs in mice. Similar to the previous study, the mice were randomly divided into an experimental and control group. The mice in the experimental group were injected with 50 μg of MSCs-exosomes via the tail vein, whereas the mice in the control group were injected with PBS via a similar route of administration. The results of this study confirmed the findings of the previous studies because the inflammation scores of the treatment group were significantly lower than the control. In addition, the pathological score was also significantly lower in the experimental group when compared with the control. Moreover, a marked reduction was also observed in the number of Th1 and Th17 cells, along with reduced proliferation of other T cell subtypes in the experimental group. Based on the findings of this study, Yongtao and colleagues (2022) decided to further investigate the potential therapeutic role of IL-10-overexpressing MSC-exosomes on EAU by using a similar experimental design as the previous study [118]. Findings of this subsequent study revealed that mice treated with IL-10 overexpressed MSC-exosomes exhibited lower Th1 and Th17 cells in the eye with increased Treg cells in the spleen and DLN. Moreover, IL-10 overexpressed MSC-exosomes more effectively suppressed the proliferation of T-cells and differentiation of Th1 and Th17 cells when compared to regular MSC-exosomes. Both preclinical studies highlighted the potential therapeutic benefits of MSC-exosomes as a novel therapy in treating EAU.

Interestingly, based on the successful penetration of MSC-exosomes into the eye, Li and colleagues (2022) explored the use of subconjunctival administration of Rapamycin loaded MSC-exosomes (Rapa-MSC exosomes) as a conjugate therapy for EAU and ocular complications caused by frequent intravitreal injections [118]. Data from the study highlighted that compared to MSC-exosomes and rapamycin alone, Rapa-MSC exosomes significantly reduced ocular inflammatory cell infiltration and protected the retinal structure of mice with EAU via similar mechanisms described in the previous studies. However, more importantly, Rapa-MSC exosomes improved the drug delivery of Rapamycin into the eye within 24 h of administration through a subconjunctival injection, highlighting the potential use of MSC-exosomes in improving drug delivery and efficacy in the eye. 

Table 2 presents a summary of the potential of MSC-derived exosomes in the treatment of posterior segment diseases and uveitis.

## 5. Overcoming Challenges in the Clinical Translation of MSC-Exosomes

While MSC-exosome therapies exhibit incredible promise, there are still several unresolved challenges: non-uniformity in the isolation and purification of MSC-exosomes, unclear treatment mechanisms of action, low-yield capacity, unstandardized large-scale production protocols, and performance characteristics—such as clinical sensitivity and specificity or sample stability—are still non-comprehensive [123]. However, what seems to be a significant obstacle to clinical transformation is the problem of exosomal product heterogeneity and the lack of standardized quality assessment criteria. Heterogeneity hinders the quality and management of MSC-exosome products, ultimately reducing reproducibility in both in vivo and in vitro contexts. The International Society for Cell and Gene Therapy has developed criteria differentiating MSCs sourced from various targets [124]. However, it needs more comprehensiveness on parental MSC quality, MSC-exosome quality, and potential for ex vivo expansion.

### 5.1. Overcoming the Hurdles of MSC-Exosome Heterogeneity

As alluded to earlier, parental and exosomal heterogeneity hinders the quality and management of MSC-exosome products, reducing their reproducibility within in vivo and in vitro contexts [125]. Naturally, this issue will produce heterogeneous results. Furthermore, published studies suggest differing parental sources will have different therapeutic effects. For instance, in the context of angiogenicity, BMSC-derived MSC-exosomes are superior to ADSC-derived by a factor of four; endometrial-derived MSC-exosomes are significantly better than both in this regard. However, ADSC-derived MSC-exomes are known to produce more cardio-protective factors such as VEGF and HGF. Correspondingly, when discussing immunomodulation, BMSC- and ADSC-derived exosomes can induce M2 polarization of macrophages; the former by a 3.2-fold increased expression of CD206 compared to the latter, which is only a 1.5-fold increase. With so much heterogeneity, more research is needed to explore the nuanced effects of different parental sources and their corresponding exosomes to determine the optimal sourcing and extraction protocol for patients.

One solution that Kou et al. (2020) suggest is extracting exosome products from human pluripotent stem cells (hPSCs)-derived MSCs [125]. The theoretical basis for this suggestion is that hPSCs would overcome the problems of source heterogeneity and, by extension, exosome product heterogeneity. The effectiveness of this approach has been studied in clinical trials exploring its use in patients with refractory graft-versus-host-disease (GVHD). While traditional MSC-EVs have shown clinical benefits for both acute and chronic GVHD, preliminary results of intravenously injected EVs extracted from hPSC-derived MSCs showed significant improvements in stiffening and dryness of the skin of cutaneous chronic GVHD patients. Analogous advantages were shown in ameliorating rejection following abdominal organ transplantation. One metric to assess cell culture quality is passage numbers; while traditional MSCs have under ten passage capacities, hPSC-MSCs were shown to have more than 30 passages. A more significant passage number indicates higher exosomal yields for clinical applications. Additionally, hPSC-MSCs are more resilient than traditional MSCs, with improved secretion and amplification abilities. These characteristics delineate higher quality MSC-exosome products and cost-efficient, large-scale production potential.

Varderidou-Minasian and Lorenowicz (2020) state that “the therapeutic efficiency of MSC therapy is not dependent on the engraftment of MSCs at the site of injury or the differentiation capability of the transplanted MSCs, but relies on their paracrine signaling” [126]. In other words, EVs, similar to exosomes, are primary determinants of therapeutic potential. For this reason, the quantitative and qualitative characteristics of MSC-secretomes should be considered in quality assessment. Recent advancements in cell culturing technology show that 3D culture systems better replicate in vivo conditions than the traditional static adherent cultures (i.e., 2D cultures). Critical traits related to MSC morphology, functionality, and structure are lost in 2D cultures; by extension, this hampers their proliferative and differentiating capacity, negatively impacting exosomal efficacy [127]. This hypothesis has been supported by Qazi et al. (2017), who demonstrated improved paracrine signaling in 3D cultures through the increased release of cytokines and growth factors [128].

Similarly, Ni Su et al. (2017) found that extracellular matrices with oriented fibers, compared to non-oriented, were more conducive to the release of anti-inflammatory and angiogenic-promoting factors [129]. With this in mind, 3D culturing can be further divided into material-free and material-supported, where the latter is more conducive for cell-to-cell connectivity and signaling [127]. Examples of material-free cell cultures include hydrogel-assisted 3D cultures and scaffold-free suspension cultures. The most notable example of material-supported cultures is hollow fiber bioreactors, demonstrating a 19.4-greater yield than 2D cultures within shorter culture periods. In addition, 3D-harvested MSC-exosomes have improved several conditions in rat models. For instance, in the context of injury repair, MSC-exosomes harvested from 3D cell cultures demonstrated improved angiogenicity as well as the proliferation and migration of endothelial cells.

Another approach to improving the quality of MSC-exosome products is through MSC preconditioning [127]. A hypoxic pretreatment involves reducing oxygen exposure, which enhances MSC proliferative capacity and genetic stability. More importantly, the upregulation of stemness genes such as SOX2 and activation of angiogenic gene transcription collectively mediate improvement in migratory and paracrine capacity. The final products are exosomes extracted from these low-oxygen tensed MSCs (Hyp-MSC-Exos). Studies investigating the therapeutic effects of these exosomes have shown benefits in various disease states, from CNS issues such as spinal cord injury to diabetic wound healing. Another pretreatment protocol called cytokine preconditioning involves the stimulation of cytokines and inflammatory factors, which is shown to increase paracrine efficiency and enhance the therapeutic potential of MSC-exosomes. In particular, exposure to TNF-α promotes the release of more inflammatory-combatting exosomes, and abundantly found in these exosomes were also inflammation-suppressing miRNAs such as miRNA-299-3p and miR-146a. Treatment with IL-1β or IFN-γ also showed the release of exosomes with increased anti-inflammatory properties and enhancement of the aforementioned miRNAs, among others. Finally, chemical and physical preconditioning approaches also exist, producing similar effects. For instance, treatment with metformin promoted more autophagy-related exosomal factors, whereas exposure to monochromatic blue light (451 nm) increased the presence of the same miRNAs as treatment with TNF-α.

### 5.2. Assessment of Parental MSCs as Proxy Indicator of MSC-Exosome Quality

A more standardized donor selection and screening approach is required for parental MSCs. High-potency MSCs are hypothesized to be more efficacious, but a complete assessment panel is required to make informed clinical decisions. New studies investigating assessment metrics involving in vitro characteristics, donor age, gender, BMI, extracellular vesicle (EV) sourcing, and genetic biomarkers have elucidated critical differentiating criteria between high- and low-quality MSCs, which can be used as proxy indicators of MSC-exosome quality.

Samsonraj et al. (2015) sought to understand the factors affecting bone marrow-derived mesenchymal stem cells (BMSCs) by delineating the relationship between in vitro characteristics and in vivo tissue regeneration potential [130]. Subcutaneous ectopic bone-forming ability was the endpoint indicator of performance. Their findings revealed that MSCs with high-efficiency colony-forming unit-fibroblasts (CFU-F) increased the number of small-sized cells with lengthened telomeres, and those with high growth capacity performed better on ectopic bone-formation assays. Specific cell-surface antigens were predictive of this performance: STRO-1⁺ MSCs and nestin⁺ MSCs expressing PDGFR-α were found in MSCs with greater capacity for high-growth and colony formation. Moreover, a global gene expression analysis revealed the genetic underpinnings of the aforementioned phenotypic differences: cellular processes in low-growth BMSCs were more maturation-based, whereas those of high-growth BMSCs were more proliferation-based.

Sathiyanathan et al. (2020) published a more detailed investigation of the genetic biomarkers of BMSCs to understand their influence on scalability [131]. The transcriptomic analysis uncovered glutathione S-transferase theta 1 (GSTT1) to be the most differentially expressed. In fact, low-growth capacity BMSCs exhibited a fifty-fold greater expression of the gene. GSTT1 was repressed in high-growth capacity BMSCs; further genotyping revealed that those donors had a genomic deletion of the GSTT1 gene. A subsequent double-blind study of the genomic influence of GSTT1 on performance indicated that GSTT1-null BMSCs demonstrated greater growth and self-renewal capacity and had lengthier telomeres. Higher total cell count and CFU-F efficiency supported this finding. They concluded that GSTT1 could be a meaningful genetic biomarker of BMSC scalability. As BMSCs demand grows, GSTT1 status in donors could be used as a rapid assessment tool for harvesting-related decisions and to inform usage in clinical applications.

Boulestreau et al. (2020) reviewed the effects of age on MSC quality [132]. Several factors contribute to the deleterious effects of senescence in the aging process. Such factors include loss of stem cell functionality or increased dysfunctionality due to age-related changes to MSCs. While the precise etiology of the relationship between MSC functionality and age remains undetermined, the following are notable agreements across the literature. First, the proliferative and clonogenic capacity of BMSCs is negatively associated with age [133]. To this end, specific cell-surface markers have been determined as meaningful indicators for MSC potency. For instance, decreased expression of CD146 has been linked to shortened telomere lengths and late-passage MSCs [134]. Similarly, the downregulation of CD106 and STRO-1, and the upregulation of CD296, have also been linked to late passage MSCs [133,135,136,137]. Third, reactive oxygen species and consequent oxidative stress are higher in aging MSCs. With this in mind, some studies have investigated possible treatments for reversing age-related changes to MSCs. For example, melatonin is protective against oxidative stress and senescence [138]. Similarly, selective inhibitor ML141 decreased CDC42 protein activity—involved in cellular division processes—in aging MSCs [139]. Siegal et al. (2013) demonstrated that the combined influence of gender and age might also affect BMSC quality [140]. They found that younger female donors, who exhibited higher clonogenicity and increased proliferative rates, had more favorable BMSCs than other donor demographics.

Ulum et al. (2018) determined that high BMI associated with obesity was a predictor of diminished MSC quality [141]. They observed a decline in critical stromal adhesion proteins and MSC markers in higher BMI donors. In these donors, BMSCs were functionally impaired, as indicated by a significant reduction in osteogenic differentiation and calcium levels, slower proliferation rates, and a higher proportion of senescent cells. The mechanism hypothesized to underlie these changes is that obesity promotes the misfolding and unfolding of proteins, increasing the expression of endoplasmic reticulum (ER) stress-related genes, specifically ATF4 and CHOP. More significant ER stress promotes stem cell dysfunction by engaging the unfolded protein response (UPR). As the global obesity rate rises, a comprehensive understanding of the challenges that obesity presents for current and future MSC donation is required. However, research has investigated potential solutions to attenuate obesity-related decreases in MSC quality. Treatment with TUDCA, a soluble bile salt used in cholestasis treatment, shows promise for this purpose. During osteogenic differentiation, TUDCA was shown to reduce the effects of ER stress and prevent UPR dysfunction. This observation was made in high-BMI BMSCs that expressed higher expression of ATF4 and CHOP. Moreover, 4-PBA—a monocarboxylic acid used to treat urea cycle disorders—regulates UPR- and ER stress-related proteins, but less significantly than TUDCA, albeit affecting its influence on different modulatory mechanisms. Both treatments increased osteogenic and adipogenic differentiation while regulating the UPR.

Li et al. (2021) comprehensively delineate modes of extracellular vesicle sourcing; namely, they highlight the benefits and drawbacks of allogeneic and autologous MSCs [142]. However, the best source remains a point of contention in the literature. The favorable safety profile and efficacy of allogeneic sourcing are reasons for its increasing use in MSC-based therapies. Its main advantages are reduced immunogenicity, high accessibility, and increased capacity for high-quality donor selection. However, similar to most allogeneic therapies, there is potential for immune rejection. Moreover, donor-donor heterogeneity may present significant challenges in selecting the best donor candidate for the disease of interest, and the patient’s system, upon infusion, quickly clears the MSCs. Autologous MSC sourcing is safer as it is sourced directly from the patient, and the MSCs themselves are immunocompromised. The primary disadvantages are long-time availability and the potential to hold disease-candidate genes. A non-trivial factor influencing the functional property of MSCs is their local microenvironment, which can harbor various safe or harmful factors. Tissue-derived MSCs from a potentially harmful microenvironment will not be favorable. Therefore, clinicians must thoroughly communicate each approach’s safety and efficacy profile to their patients to optimize treatment outcomes and patient satisfaction.

## 6. Conclusions

In conclusion, the use of mesenchymal stem cell (MSC) derived exosomes in ophthalmology has garnered significant attention due to their ability to circumvent the limitations of traditional MSC-based treatments. MSC-exosomes have unique properties, such as the ability to rapidly pass through biological barriers, deliver immunomodulatory and trophic factors, and avoid unwanted differentiation and immunological rejection. This article sheds light on the distinctive features and biological foundations of MSC-derived exosomes while conducting a thorough examination of preclinical studies that have demonstrated the potential of these exosomes in treating various ocular diseases from the front to the back of the eye.

Despite challenges and limitations that require addressing in preclinical studies, recent advances in MSC-derived exosomes hold great promise for the future. By overcoming these challenges, the use of MSC-derived exosomes could offer a safer and more effective alternative to MSC-based therapies, revolutionizing our collective approach to the management of ophthalmic diseases and bringing clearer visions and a brighter future for patients.

## Figures and Tables

**Figure 1 pharmaceutics-15-01167-f001:**
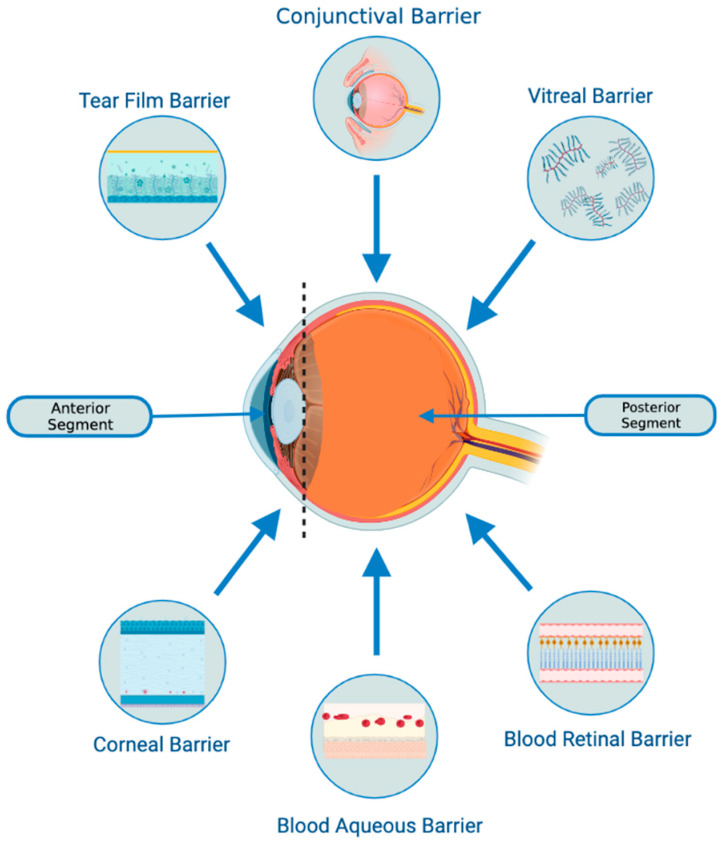
Biological barriers of the eye. The tear film barrier comprises three layers (lipid, aqueous, and mucin) and is continuously removed from the eye surface by lacrimal fluid secretion, resulting in rapid drug removal. The corneal barrier serves as a mechanical and chemical barrier, limiting the access of exogenous substances into the eye. Tight junctions on the corneal epithelium surface prevent the diffusion of macromolecular and hydrophilic molecules. Intravitreal administration offers a direct path to the vitreous and retina but may impede the diffusion of larger, positively charged drugs across the retinal pigment epithelium (RPE) barrier to the choroid. The blood-ocular barrier (BOB) poses a significant challenge to systemic and topical drug delivery in the anterior and posterior chambers of the eye. It comprises the blood-aqueous barrier (BAB) and the blood-retinal barrier (BRB). The BAB, associated with the anterior chamber, consists of endothelial cells, iris, ciliary muscle, and pigmented and non-pigmented epithelium cells with tight junctions that restrict drug molecule entry.

**Figure 2 pharmaceutics-15-01167-f002:**
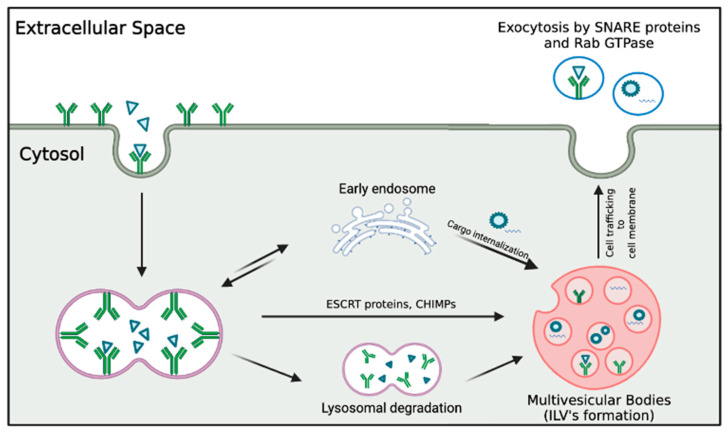
Features of exosome biogenesis. Exosome formation begins with the endocytosis of cell membrane components and lipids, initiating the formation of early endosomes in the cytosol. These endosomes undergo sorting by ESCRT and CHMP proteins, resulting in the formation of multivesicular bodies that house intraluminal vesicles. These multivesicular bodies are then trafficked to the cell membrane, where the exocytic pathway releases exosomes into the extracellular space.

**Figure 3 pharmaceutics-15-01167-f003:**
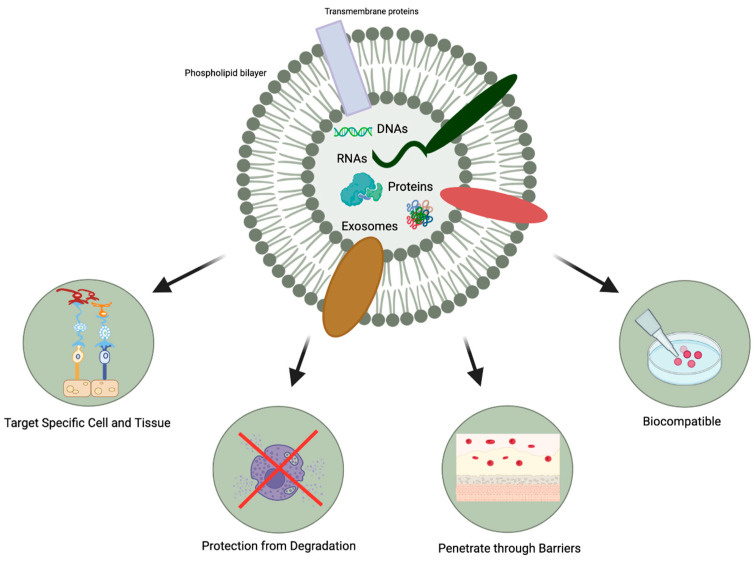
Favorable characteristics of MSC-exosomes.

**Table 1 pharmaceutics-15-01167-t001:** MSC-derived exosomes in anterior segment diseases and glaucoma.

Disease	Origin	Route of Administration	Results	Stage	Reference
Corneal wound	Human corneal MSC-derived exosomes	Topical	- Significantly greater healing in treated wounds - Re-epithelialization significantly accelerated in monolayers of confluent human corneal epithelial cells in culture (*p* < 0.005) - Acceleration of murine corneal epithelial wound healing in vivo (*p* < 0.05)	Preclinical trial (in vitro and in vivo mouse study)	[18]
Corneal alkali injury	Human placenta-derived MSC EVs	Topical	- Promotion of corneal epithelial migration (dose-dependent pattern; Ki-67% increased), with significant improvement in wound healing - Inhibition of inflammation and apoptosis (decrease in RNA levels of IL-1b, IL-8, TNF-a, NF-kB both in vitro + in vivo, IL-10 increased in vitro, decrease in Cas-8) - Limited neovascularization and corneal hazing (decrease in MMP-2 and MMP-9, VEGF)	Preclinical trial (in vitro and in vivo mouse study)	[48]
Corneal wound	Adipocyte-derived MSC-derived exosomes	Topical	- Increase in growth of corneal stromal cells and reduced rate of apoptosis - Downregulation of MMP expression (MMP1, MMP2, MMP3, MMP9), upregulation in collagen synthesis (promotion of ECM synthesis) - Growth of corneal stromal cells increased in a dose-dependent manner with exosome concentration - Inhibit apoptosis of corneal stromal cells	Preclinical trial (in vitro)	[52]
Corneal wound	Bone marrow derives MSC-exosomes	Topical	- Significant promotion of corneal cell migration - Reduced expression of MMP-2 - Potential inhibitory effect on inflammation and neovascularisation - Mild corneal epithelial scratch wound - Association between MMP-2 and angiogenesis	Preclinical trial (in vitro, “cornea on a chip”: combination of human corneal cells and microfluidics to mimic ocular surface in vitro)	[49]
Corneal wound	Human umbilical cord MSC-derived exosomes combined with autophagy activator (AA) Rapamycin	Topical	- Increased autophagic activity - 50 nM Rapamycin showed positive effects in cell viability and migration acceleration - Exo + AA better than either alone - Apoptosis percentage lowest in Exo + AA group, highest in Exo + AI - Reduced haze grade - Promotion of cellular proliferation - Samples treated with human umbilical MSC-derived exosomes alone showed increased proliferation and reduced hazing, and these effects were increased when AA was added - Downregulation TNF-a, IL-1b, IL-6, CXCL-2 in mice	Preclinical trial (mouse model)	[51]
Corneal wound	Adipose-derived MSC-derived exosomes loaded with miRNA 24-3p in thermosensitive hyaluronic acid hydrogel (THH)	Topical(miRNA 24-3p was tested alone by subconjunctival injection)	- Limited post-operative corneal edema - Recovery of normal corneal thickness but poor morphology and lack of tight junctions (miRNA 24-3p alone had healthy epithelium but abnormal thickness) - Positive influence on corneal epithelial cell migration	Preclinical trial (rabbit study)	[58]
Corneal wound	Induced pluripotent stem cell-derived MSC (iPSC-MSC) exosomes containing miR-432-5p (in a thermosensitive hydrogel for part of the study)	Topical	- Accelerated closure with improved migration - Downregulation of collagen expression in corneal stromal stem cells (with dramatic upregulation when miR-432-5p inhibitor was added) - Downregulation of a-SMA expression and vimentin - miR-432-5p potential effect on corneal transparency and reduced scarring	Preclinical trial (in vitro and in vivo rat study)	[57]
Corneal wound	Human umbilical MSC-derived exosomes with miR-21	Topical	- Upregulation of migration and proliferation - This upregulation was partially negated by miR-21 knockdown - Action through regulation of PTEN/PI3K/Akt pathway - Downregulation of PTEN in miR-21 treated samples - Posttranscriptional modification - HCECs in culture - Promote healing of corneal defects in rats, more regular arrangement and compact structure	Preclinical trial (in vitro and in vivo rat model)	[50]
GVHD-associated dry eye disease	Mouse bone marrow-derived MSC-derived exosomes (mouse study), human umbilical cord MSC-derived exosomes (clinical study)	Topical	- miR-204 targets IL-6R to induce shift of M1 macrophages to immunosuppressive M2 phenotype - Increased tear secretion and reduced corneal edema and hazing were seen in mouse models - MSC-exosome treatment prevented corneal degeneration, increased the thickness of the central cornea and epithelium, restored corneal structure - Downregulation of pro-inflammatory genes - Protective effect in conjunctiva and lacrimal gland - miR-204 knockdown exacerbated DED	Preclinical trial (2 dry eye mouse models) and clinical study (14 patients/28 eyes with GVHD)	[4]
GVHD-associated DED	Human umbilical MSC-derived exosomes	Topical	- Estimated completion by 2023	Clinical trial (phase II) with 27 study subjects affected by dry eye symptoms with cGVHD	[67]
DED	Human adipose-derived MSC-EVs	Topical	- Marked decrease in desiccating stress in treated eyes - No difference in normal eyes treated with human adipose-derived MSC-EVs, showing the safety of the treatment - Inhibition of cell apoptosis - Suppression of NLRP3 activation-mediated inflammation	Preclinical trial (mouse model)	[65]
DED	MSC-derived EVs	Topical	- Increased tear production - Upregulation of dendritic cells in DED, and treatment effectively reduced dendritic cell count, suppressed expression of MHC-II - Downregulation of inflammatory cytokines	Preclinical trial (in vitro and in vivo mouse study)	[63]
DED	Mouse adipose-derived MSC-derived exosomes	Topical	- Preservation of goblet cells in the conjunctiva - Reduction in apoptosis - Suppression of inflammatory cytokines, stimulation of anti-inflammatory cytokine IL-10 - Suppression of NLRP3 inflammasome activation	Preclinical trial (mouse model)	[64]
DED	Mouse MSC-derived exosomes coupled with ascorbic acid	Topical	- Percentage of M2 macrophages increased by ascorbic acid, which reduces inflammation by removal of ROS, decreasing hyperosmolarity of the tear film - Treatment that optimized the minimal corneal damage, increased thickness of the central cornea, and restoration of the stroma layer - Significant increase in tear secretion	Preclinical trial (mouse model)	[66]
DED	MSC-exosomes	Topical	- Epithelium ultrastructure improved with more corneal chondriosome/desmosomes, better morphological features of microvilli - More tear production, longer tear break-up time	Preclinical trial (mouse model)	[95]
cGVHD	Human bone marrow-derived MSC-exosomes	Tail vein injection	- Suppression of CD4 cells, TH17 cells - Upregulation of IL-10-expressing cells - Reduced pro-inflammatory cytokine production	Preclinical study	[61]
SSDE	Human umbilical-derived MSC-exosomes	In vitro peripheral blood mononuclear cells (PBMCs)	- Restore the balance in mi-RNA-125b-5p and miRNA-155-5p expression in CD4+ T cells - Inhibition of T cell proliferation and activation	Preclinical study (in vitro)	[69]
SSDE	Labial gland-derived MSC-exosomes	Tail vein injections	- Induction of Tregs and suppression of Th17	Preclinical study (mouse model)	[70]
SSDE	Olfactory ecto-MSC-exosomes	Intravenous injections	- Secretion of IL-6 from olfactory ecto-MSC-exosomes and activation of myeloid-derived suppressor cells (MDSC) through Stat3 pathway		[72]
Mucopolysaccharidosis IVA	Human umbilical MSC-derived EVs		- Human umbilical MSC-EVs deliver functional GALNS enzyme to deficient cells	Preclinical trial (in vitro)	[75]
Glaucoma/optic nerve crush	Bone marrow-derived MSC-exosomes	Intravitreal injection, just posterior to the limbus	- Improved neuritogenesis - Reduction of RGC loss after 21 days from 80–90% to 30% in the BMSC-exo treated sample - Significant neuroprotection and preservation of function - Dependency on the miRNA cargo in exosomes shown by AGO2 knockdown, with important candidates miR-21, miR-146a, miR-17-92	Preclinical study (rat study)	[77]
Glaucoma/optic nerve crush	Human umbilical MSC-exosomes	Intravitreal injection	- Neuroprotective effect but no axogenesis, likely due to miRNA content in different MSCs (bone marrow MSC vs. human umbilical MSC) - Effect has shown to be related to miRNA content of exosomes through Argonaute-2 knockdown - Activation of glial cells with the possible secretion of neurotrophins	Preclinical trial (rat study)	[30]
Glaucoma/ONC	miR-21	Intravitreal injection	- Attenuation of astrocyte activation - Excessive activation of glial cells possibly contributes to RGC degeneration - Promotion of axogenesis and functional recovery in injured optic nerve	Preclinical trial (rat study)	[87]
Glaucoma (POAG)	Bone marrow-derived MSC-exosomes		- Improved human trabecular meshwork cell viability after exposure to hydrogen peroxide - Reduced production of iROS after exposure - Downregulation of proinflammatory factors - Upregulation of MMP2 and MMP3 - Potential alleviation of human trabecular meshwork cell dysfunction induced by oxidative stress - Action through miRNA, with downregulation of miR-126-5p in the exosome group (this miRNA is upregulated in tears of glaucoma patients) - Action of miR-3529-3p may reduce the inflammatory response to oxidative stress by acting on CXCL5	Preclinical trial (in vitro)	[91]
Glaucoma/ONC	Human embryonic stem cell-MSC-EVs	Tail vein injections	- Significant protection of retinal ganglion cell axons - Suppression of cis p-tau accumulation, an early driver of tauopathy and neurodegeneration process	Preclinical trial (mouse model)	[29]
Glaucoma/ONC	Human bone marrow-derived MSC-exosomes		- Priming of MSCs with TNF-a significantly increased the neuroprotective effect of the MSC-exosome treatment on rat and human retinal ganglion cells in culture - Non-primed MSC-exosomes have a neuroprotective effect, but the effect is more muted without priming - Priming has no effect on the division rate or secretory rate of exosomes, with a significant positive influence on neurotrophic factor expression (VEGF, HGF, IGF)	Preclinical trial (in vitro)	[80]
Glaucoma	Human bone marrow MSC-derived exosomes	Intravitreal injection	- Neuroprotection of retinal ganglion cells from death, with a decrease in the number of degenerating axons	Preclinical study (mouse model)	[78]
Glaucoma	Human bone marrow MSC-derived exosomes	Intravitreal injection	- AGO2 knockdown (depletion of miRNA) partially inhibited the positive effects of the exosome treatment - Promotion of neuroprotection and functional preservation of retinal ganglion cells in 2 rat glaucomatous models - No direct effect on intraocular pressure (IOP), but the potential to be used as adjunctive therapy to IOP-lowering medications	Preclinical study (rat model)	[79]
Optic nerve injury	Human placenta-derived mesenchymal stem cells (hPSCs)	In vitro immortalized R28 retinal precursor cells exposure	- Restored cell proliferation and mitochondrial quality control in R28-damaged cells	Preclinical study (in vitro)	[81]

**Table 2 pharmaceutics-15-01167-t002:** MSC-derived exosomes in posterior segment diseases and uveitis.

Disease	Origin	Route of Administration	Results	Stage	Reference
Idiopathic Macular Holes	Human umbilical cord-derived mesenchymal stem cells (hUCMSCs)	Intravitreal injection	MSC-Exo therapy promotes functional and anatomic recovery from MH and can be used as a. safe method for improving the visual outcomes after MHs surgery.	Clinical Trial	[112]
Retinitis Pigmentosa	Mouse bone marrow MSC	Intravitreal Injection	Intravitreal MSCT counteracted photoreceptor apoptosis and alleviated retinal morphological and functional degeneration in a mouse model of photoreceptor loss.	Preclinical Study	[97]
Retinitis Pigmentosa	Mouse bone marrow MSC	Intravitreal Injection	MSC-exosomes might suppress the degeneration of photoreceptors with increased thickness of the retinal outer nuclear layer, and mechanically, MSC-exosome treatment inhibits the expression of pro-inflammatory cytokines, indicating the relief of neuroinflammation.	Preclinical Study	[119]
Retinitis Pigmentosa	Human umbilical cord mesenchymal stem cells-derived exosome	Intravitreal Injection	MSC-Exosomes increased the survival of photoreceptors and preserved their structure. Visual function, as reflected by optomotor and electroretinogram responses, was significantly enhanced in MSC-EVs-treated rd10 mice.	Preclinical Study	[98]
Diabetic Retinopathy	Mice Bone marrow-derived Mesenchymal stem cell (BMSC)	In Vitro culturing of Mouse retinal cells (Muller cells) with MSCs-exosome was performed	BMSC-derived exosomes inhibited oxidative stress, inflammation, and apoptosis and promoted the proliferation of HG-treated Muller cells.	Preclinical Study	[100]
Diabetic Retinopathy	Human umbilical cord mesenchymal stem cells-derived exosome	Intravitreal Injection	hucMSCs-derived exosomes shuffle miR-17-3p to ameliorate inflammatory reaction and oxidative injury of DR mice via targeting STAT1.	Preclinical Study	[101]
Diabetic Retinopathy	Bone marrow mesenchymal stem cell (BMSC)	Injection into the Retinal Ganglion Cell Culture of DR model mice was conducted ex vivo	BMSC exosomal miR-146a can regulate the inflammatory response of DR by mediating the TLR4/MyD88/NF-κB pathway, providing an experimental basis for the prevention and treatment of DR.	Preclinical Study	[102]
Diabetic Retinopathy	Human umbilical cord-derived mesenchymal stem cells (hUCMSCs)	Intravitreal Injection	miR-126 expression in MSC-Exos reduces hyperglycemia-induced retinal inflammation by downregulating the HMGB1 signaling pathway.	Preclinical Study	[7]
Diabetic Retinopathy	Animal (Rabbit) adipose-derived mesenchymal stem cells	Intravenous Subconjunctival and Intraocular	Increased expression of micRNA-222 with regenerative changes in the retina following administration of MSCs-derived exosomes.	Preclinical Study	[99]
Diabetes Retinopathy	Bone marrow-derived mesenchymal stem cell-derived exosomes (BM-MSCs-Ex)	Intravitreal Injection	By blocking the wnt/b-catenin pathway in the diabetic retina, exosomes demonstrated a significant reduction in features of DR.	Preclinical Study	[103]
Retinal Ischemia	Human Bone marrow mesenchymal stem cell (HuBMSC)	Intravitreal	hMSCs were well tolerated without immunosuppression and decreased the severity of retinal ischemia in this murine model.	Preclinical Study	[38]
Retinal ischemia-reperfusion injury	Gingival MSC (GMSC)-exosomes	Intraocular	GMSC exosomes significantly reduced inflammation and retinal cell loss caused by glaucoma without any autoimmunity.	Preclinical Study	[108]
Retinal Injury/ Ischemia	Mouse adipose and human umbilical cord mesenchymal stem cell	Intravitreal In vivo and in vitro Retinal cell	MSCs and their exosomes reduced damage inhibited apoptosis, and suppressed inflammatory responses to obtain a better visual function to nearly the same extent in vivo.	Preclinical Study	[4]
Retinal Detachment	Sprague-Dawley rat bone marrow-derived mesenchymal stem cell exosomes	Subretinal Injection	Suppression of photoreceptor cell apoptosis and maintenance of normal retinal structure observed when treated with MSC-Exosomes.	Preclinical Study	[109]
Age-related macular degeneration	Retinal astroglial cell-derived exosomes	Intravitreal	Retinal Astroglial cell-derived exosomes inhibited laser-induced choroidal neo-vascularization.	Preclinical Study	[105]
Age-related macular degeneration	Human umbilical cord-derived mesenchymal stem cells (hUCMSCs)	Intravitreal injection in vivo and Retinal pigment epithelial cell model in vitro	In vivo, MSCs-derived exosomes reduced damage, distinctly downregulated VEGF-A, and gradually improved the histological structures of CNV for better visual function. In vitro, MSCs-derived exosomes downregulated the mRNA and protein expression of VEGF-A in RPE cells after blue light stimulation.	Preclinical Study	[106]
Subretinal fibrosis resulting from neovascular age-related macular degeneration	Human umbilical cord-derived mesenchymal stem cells (hUCMSCs)	Intravitreal injection	Intravitreal injection of hucMSC-Exo effectively ameliorated laser-induced CNV and subretinal fibrosis via the suppression of epithelial–mesenchymal transition (EMT) process.	Preclinical Study	[120]
Experimental Autoimmune Uveitis	Human bone marrow mesenchymal stem/stromal cells (huBMSCs)	Intraperitoneal	hMSCs attenuate EAU by suppressing Th1/Th17 cells and induce IL-10-expressing B220+CD19+ cells.	Preclinical Study	[121]
Type 1 Diabetes and Experimental Autoimmune Uveitis	Human Bone marrow mesenchymal stem cells (HuBMSCs)	Tail vein intravenous injection	huBMSC-exosomes effectively prevent the onset of disease in both T1D and EAU.	Preclinical Study	[115]
Experimental Autoimmune Uveitis	Human umbilical cord-derived mesenchymal stem cells (hUCMSCs	Periocular Injection	MSC-exosomes effectively ameliorate EAU by inhibiting the migration of inflammatory cells, indicating a potential novel therapy of MSC-Exo for uveitis.	Preclinical Study	[116]
Experimental Autoimmune Uveitis	Rat mesenchymal stem cell	Periocular Injection	MSC-exosomes can reduce the clinical and pathological manifestations of EAU, protect retinal function, reduce ocular macrophage infiltration, down-regulate the proportion of inflammatory cells in the eye, and inhibit T cell proliferation.	Preclinical Study	[117]
Experimental Autoimmune Uveitis	Human umbilical cord-derived mesenchymal stem cells (hUCMSCs)	Intravenous Injection	Intravenous injection of human umbilical cord MSCs-derived sEVs can reduce inflammation in EAU mice.	Preclinical Study	[118]
Experimental Autoimmune Uveitis	Human umbilical cord-derived mesenchymal stem cells (hUCMSCs)	Intravenous Injection (tail vein)	IL-10 overexpressed MSC-exosomes effectively ameliorate EAU by regulating the proliferation and differentiation of T-cells.	Preclinical Study	[118]
Experimental Autoimmune Uveitis	Human umbilical cord-derived mesenchymal stem cells (hUCMSCs)	Subconjunctival injection	Compared to sEVs and rapamycin alone, Rapa-sEVs can produce a more marked therapeutic effect and reduce ocular inflammatory cell infiltration.	Preclinical Study	[122]
Diabetes Retinopathy	Human Bone marrow mesenchymal stem cell (HuBMSC)	Cellular exposure ex vivo on Human retinal microvascular endothelial cells	MSC-derived exosomal lncRNA SNHG7 suppresses endothelial–mesenchymal transition and tube formation in Human retinal microvascular endothelial cells.	In Vitro Preclinical Study	[104]
Retinal Ischemia	Human Bone marrow mesenchymal stem cell (HuBMSC)	Cellular exposure ex vivo on Human retinal microvascular endothelial cells	Retinal cells exposed to MSC-derived EVs significantly reduced cell death and attenuated loss of cell proliferation.	In Vitro Preclinical Study	[107]

## Data Availability

Not applicable.

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
