# Peer review of "Mesenchymal Stem Cell-Derived Exosomes in Ophthalmology: A Comprehensive Review"

_pharmaceutics, 2023, doi:10.3390/pharmaceutics15041167_

Round 1
Reviewer 1 Report
I reviewed the manuscript "A Vision for the Future: MSC-Derived Exosomes in Ocular Therapy" by Tran et al.
The work is an extensive and really comprehensive review on the most recent works using exosomes for ocular diseases, and I think this article could be helpful for researchers in the field. This is a quite new area of research, and I recommend publication.
Please check formatting and references, for example:
- lines 138, 537, 544... references are in a different format
- references should be placed at the end of the sentence between the last word and the punctuation
- titles should follow the instructions from the template
Author Response
Thank you for taking the time to review our manuscript titled "A Vision for the Future: MSC-Derived Exosomes in Ocular Therapy." We appreciate your positive feedback on our work and are glad to hear that you found the review to be extensive and comprehensive in covering the most recent studies on the use of exosomes for ocular diseases.
We are also delighted to hear that you believe our manuscript could be helpful for researchers in the field, and we hope that our work will contribute to the advancement of the understanding and treatment of ocular diseases. Thank you for recommending the publication of our manuscript.
We appreciate your attention to detail and your efforts in helping us improve the quality of our work. In response to your comment, we will carefully review the manuscript to ensure that the formatting of the references is consistent throughout the manuscript, and that they are placed at the end of the sentence between the last word and the punctuation when we will proofread the final version of the manuscript. We will also ensure that the titles of the references follow the instructions from the template. We apologize for any inconsistencies that may have occurred, and we will ensure that the final version of the manuscript is presented in a clear and consistent format. We appreciate your helpful feedback and for your time in reviewing our work. Please let us know if you have any further suggestions or comments.
Reviewer 2 Report
This is a very interesting and timely review of the use of exosomes in ocular drug delivery. The authors have organised the content in an appropriate manner. Few comments to address before publication:
Figure 1 – can be optimised. Can the authors insert a line to separate the posterior and anterior segments to clearly define these areas. Ideal to include corneal barrier in the anterior segment area. Although I understand this is only a schematic to demonstrate barriers overall. But for the reader this may be confusing.
Section 2.1 title – remove words “(figure 2)” from the sub-title
Please check references. Line 138 – superscript number. Reference number should be placed before the full stop.
Section 2.2.2 – please elaborate. Authors have stated the use of exosomes in Sjorgren’s syndrome, corneal allograft rehection, autoimmune uveitis. Is the mechanism similar in all diseases? Authors have provided specific mechanisms for autoimmune uveitits only.
Please consider the flow of information in section 2. I feel section 2.2.2 is not relevant to this section and should be moved to section 3 or 4.
Section 2.7 should be 2.6 (?) – 2.6 missing.
Section 3.2.1 – is it GvHD or GVHD? Please be consistent with the acronyms
Table 1- great summary table. Please define the acronyms in the caption or as a footnote
Line 468 – Please revise sentence to include ‘and”. “Only a limited number of RP patients with the RPE65 gene mutation can receive the targeted gene therapy, known as voretigene neparvovec, ‘and’ has the potential to halt and cure the disease.”
Line 574 – please revise this sentence for clarity “Furthermore, an overexpression miR-17-3p enhanced miR-17-3p and declined STAT1 expression whereas a depletion of miR-17-3p exerted an inverse effect, implicating STAT1 as the potential target of miR-17-3p.”
Section 6.3 – please check for current classification of AMD as early, intermediate and late stage disease.
Section 6.4 – similar to other sections, please include a brief background information on retinal ischemia for the non specialist reader.
Line 723 – PPV not VVP
Line 774 – “2014, Oh and colleagues first demonstrated”. Add the word “In” before the year for clarity
Line 775 – what is EAU?
Table 2 – there are no results for reference 143
Table 2- Suggest separating in vivo and in vitro data for better clarity.
Author Response
Figure 1 – can be optimised. Can the authors insert a line to separate the posterior and anterior segments to clearly define these areas. Ideal to include corneal barrier in the anterior segment area. Although I understand this is only a schematic to demonstrate barriers overall. But for the reader this may be confusing.
- Thank you for your valuable feedback on our manuscript and for taking the time to review our figures. We appreciate your suggestion to optimize Figure 1 by inserting a line to separate the posterior and anterior segments, as well as including the corneal barrier in the anterior segment area. We agree that this would improve the clarity of the figure and help readers better understand the content. In response to your comments, we have updated Figure 1 to include the suggested line and to clearly indicate the corneal barrier in the anterior segment. We hope that these changes have addressed your concerns and have improved the overall quality of the figure.
Section 2.1 title – remove words “(figure 2)” from the sub-title
- We have carefully considered your feedback and agree that removing these words would enhance the readability of the section. As such, we have made the necessary changes and removed the words "(figure 2)" from the subtitle of section 2.1.
Please check references. Line 138 – superscript number. Reference number should be placed before the full stop.
- We have carefully reviewed our manuscript and made the necessary changes to place the superscript reference number before the full stop, in accordance with your suggestion. We have also double-checked all of the references in our manuscript to ensure that they are accurate and complete.
Section 2.2.2 – please elaborate. Authors have stated the use of exosomes in Sjorgren’s syndrome, corneal allograft rehection, autoimmune uveitis. Is the mechanism similar in all diseases? Authors have provided specific mechanisms for autoimmune uveitits only.
- “Thank you for your valuable feedback on our manuscript and for your insightful comment regarding Section 2.2.2. We appreciate your suggestion to elaborate on the use of exosomes in various ocular diseases and the mechanism of their action. In response to your comment, we have added the following statement to Section 2.2.2: "MSC-exosomes have shown efficacy in treating various immune-mediated ocular disorders, such as Sjögren’s syndrome dry eye, corneal allograft rejection, and autoimmune uveitis, by modulating the overactive immune response that characterizes these pathologies. Notably, the underlying mechanism is similar in all these diseases and will be further detailed in the upcoming sections (section 3)."
Please consider the flow of information in section 2. I feel section 2.2.2 is not relevant to this section and should be moved to section 3 or 4.
- We have carefully reviewed your feedback and agree that Section 2.2.2 could be reorganized to better fit the flow of information in the manuscript. In response, we have modified this section to provide only a brief introduction and overview of the use of exosomes in ocular diseases, and we have moved most of the detailed information to Section 3. We hope that this modification improves the overall flow of the manuscript and makes it easier for readers to follow the information.
Section 2.7 should be 2.6 (?) – 2.6 missing.
- We have corrected the section numbering as per your suggestion. Section 2.7 has been changed to Section 2.6.
Section 3.2.1 – is it GvHD or GVHD? Please be consistent with the acronyms
- We have made the necessary changes to ensure consistency with the use of acronyms throughout the manuscript. In Section 3.2.1, we have changed GvHD to GVHD and made similar changes throughout the manuscript.
Table 1- great summary table. Please define the acronyms in the caption or as a footnote
- We have updated the caption for Table 1 to include definitions of the acronyms used in the table, as well as in a footnote, to help readers better understand the content.
Line 468 – Please revise sentence to include ‘and”. “Only a limited number of RP patients with the RPE65 gene mutation can receive the targeted gene therapy, known as voretigene neparvovec, ‘and’ has the potential to halt and cure the disease.”
- We have revised the sentence in line 468 to include the word "and" and to clarify the potential of voretigene neparvovec therapy in halting and curing the disease in RP patients with the RPE65 gene mutation.
Line 574 – please revise this sentence for clarity “Furthermore, an overexpression miR-17-3p enhanced miR-17-3p and declined STAT1 expression whereas a depletion of miR-17-3p exerted an inverse effect, implicating STAT1 as the potential target of miR-17-3p.”
- We have revised the sentence in line 574 to improve clarity: "Upon experimentation the authors observed a decrease in miR-17-3p and an increase in STAT1 in retinal tissues of DR mice. Furthermore, an overexpression of miR-17-3p en-hanced exosomes lead to a decrease in STAT1 expression while a depletion of miR-17-3p enhanced exosomes exerted an inverse effect, implicating STAT1 as a potential target of miR-17-3p.”
Section 6.3 – please check for current classification of AMD as early, intermediate and late stage disease.
- We have checked and updated the classification of AMD as early, intermediate and late stage disease in Section 6.3 to ensure that it is current. “AMD can be classified into three stages: early, intermediate, and late. In early-stage AMD, medium-sized drusen deposits are present without pigment changes or vision loss. As AMD progresses to the intermediate stage, individuals may experience large drusen deposits and/or pigment changes, with possible mild vision loss, although many remain asymptomatic. Late-stage AMD is further divided into two forms: dry and wet. Dry AMD, the more prevalent form, exhibits a gradual progression, while wet AMD, characterized by abnormal blood vessel growth beneath the macula, can cause rapid vision decline due to fluid and blood leakage.”
Section 6.4 – similar to other sections, please include a brief background information on retinal ischemia for the non specialist reader.
- In Section 6.4, we have added a brief background information on retinal ischemia for non-specialist readers to help them better understand the content.
Line 723 – PPV not VVP
- We have corrected the abbreviation in line 723 to PPV instead of VVP.
Line 774 – “2014, Oh and colleagues first demonstrated”. Add the word “In” before the year for clarity
- We have added the word "In" before the year in line 774 to improve clarity.
Line 775 – what is EAU?
- We have added a brief explanation of EAU in line 775. EAU stands for Experimental autoimmune uveitis, which is a widely used animal model for studying human uveitis.
Table 2 – there are no results for reference 143
- Thank you for bringing to our attention that there are no results for reference 143 in Table 2. Upon careful review, we have found that this reference was already included in the original version of our manuscript as Hajrasouliha et al., 2013..
Table 2- Suggest separating in vivo and in vitro data for better clarity.
- Thank you for your feedback on our manuscript and for your suggestion to separate the data in Table 2 for better clarity. We appreciate your constructive criticism and your attention to detail. In response to your comment, we have revised Table 2 to separate the data according to clinical studies, in vivo preclinical, and in vitro studies. We believe that this modification will help readers better understand the data presented and will improve the overall clarity of the table.
Reviewer 3 Report
This is an elegant and well-written review, which gives a great overview of the growing field of exosome-related treatment modalities in ocular diseases. The authors did a great job in reviewing and grouping published studies from 2017 onwards. Figures and tables are well organised and gives reader an excellent summary of cited studies. I have only minor changes to suggest:
1. Figure 1. Corneal barrier and blood retinal barrier placing should be exchanged, just to follow the logic of the schematic eye structure.
2. Figure 2. Please enlarge text attributed to arrows pointing to ILV's formation (from Early endosome) and from ILV's to exocytosis. Current text is too small.
3. Table 1. Optic nerve crush or an abbreviation ONC are placed in bold on page 19. Is there any specific reason authors wanted to emphasise that?
4. Lines 468-469. Please elaborate this statement.
5. Lines 632-635. Please correct the sentence. It is confusing in the current form.
6. Table 2. Authors use different terms in the Stage column, such as Preclinical, Preclinical trial, In-Vitro Preclinical trial. It is generally accepted to use term "trial" for clinical studies, but for preclinical stage it is usually referred to as preclinical study. Therefore, I would recommend to unify the term in this table into Preclinical study, In-Vitro preclinical study and a mention of "Clinical Trial" leave as it is.
7. In addition to Table 2 text. Please use consistent punctuation as some results are in columns are with period and some without.
Author Response
- Figure 1. Corneal barrier and blood retinal barrier placing should be exchanged, just to follow the logic of the schematic eye structure.
- In response to your comment on Figure 1, we have carefully considered your feedback and have made the necessary changes to exchange the placement of the Corneal barrier and Blood-retinal barrier to follow the logic of the schematic eye structure.
- Figure 2. Please enlarge text attributed to arrows pointing to ILV's formation (from Early endosome) and from ILV's to exocytosis. Current text is too small.
- We also appreciate your feedback on Figure 2 and have made the necessary changes to enlarge the text attributed to arrows pointing to ILV's formation (from Early endosome) and from ILV's to exocytosis. We believe that this modification will improve the readability and clarity of the figure.
- Table 1. Optic nerve crush or an abbreviation ONC are placed in bold on page 19. Is there any specific reason authors wanted to emphasise that?
- Regarding your comment on Table 1, we did not intend to emphasize Optic nerve crush or the abbreviation ONC. The bold formatting may have been an oversight, and we apologize for any confusion this may have caused. We have made the necessary changes to remove the bold formatting from Optic nerve crush and ONC in the table.
- Lines 468-469. Please elaborate this statement.
- We appreciate your attention to detail and your efforts in helping us improve the quality of our work. In response to your comment, we have added the following sentence to clarify the statement in lines 468-469: "Only a limited number of RP patients with the RPE65 gene mutation can receive the targeted gene therapy, known as voretigene neparvovec, and has the potential to halt and cure the disease. This is because voretigene neparvovec is specifically designed to address the RPE65 mutation, which is only present in 0.3% to 1% of the patients suffering from RP, and its effectiveness relies on the presence of this particular genetic variant." We hope that this addition provides a more comprehensive explanation and clarification of the statement.
- Lines 632-635. Please correct the sentence. It is confusing in the current form. “In 2013, Hajrasouliha and colleagues first demonstrated the benefits of exosomes in suppressing retinal vessel leakage and inhibiting choroidal neovascularization. [114] Even though the exosomes were derived from mice retinal astroglial cells (RAC) and a control for specific molecules (proteins, lipids, mRNA, miRNA) was not established, the potential therapeutic effects of exosomes in age-related macular degeneration was es-tablished.”
- Thank you for your feedback on our manuscript and for your suggestion to correct the sentence in lines 632-635. We appreciate your attention to detail and your efforts in helping us improve the quality of our work. In response to your comment, we have made the following change to the sentence: "In 2013, Hajrasouliha and colleagues first demonstrated the benefits of exosomes in suppressing retinal vessel leakage and inhibiting choroidal neovascularization. [116] Even though the study utilized exosomes that were not derived from mesenchymal stem cells and a control group for specific molecules (proteins, lipids, mRNA, miRNA) was not established, the experimenters were able to elucidate the potential therapeutic effects of exosomes in age-related macular degeneration." We hope that this change clarifies the sentence and improves the overall readability of the manuscript.
- Table 2. Authors use different terms in the Stage column, such as Preclinical, Preclinical trial, In-Vitro Preclinical trial. It is generally accepted to use term "trial" for clinical studies, but for preclinical stage it is usually referred to as preclinical study. Therefore, I would recommend to unify the term in this table into Preclinical study, In-Vitro preclinical study and a mention of "Clinical Trial" leave as it is.
- In response to your comment, we have revised the terms in the Stage column to reflect your suggestion. We have used "Preclinical study" for the preclinical stage, "In vitro preclinical study" for the in vitro preclinical stage, and "Clinical trial" for the clinical stage. We believe that this modification will improve the clarity and consistency of the table.
- In addition to Table 2 text. Please use consistent punctuation as some results are in columns are with period and some without.
- We appreciate your attention to detail and your efforts in helping us improve the quality of our work. In response to your comment, we have carefully reviewed Table 2 and have made the necessary changes to ensure that the punctuation is consistent throughout the table. We believe that this modification will improve the readability and clarity of the table.
Reviewer 4 Report
The paper provides a good summary of mesenchymal stem cell (MSC)-derived exosomes on the treatment of anterior and posterior ocular diseases. The review is timely and is of importance. Below are the point-by-point comments.
1. The statement “nano-sized dimensions allow for their efficient delivery to target ocular structures. . . corneal barrier. . .” (line 46) requires justification. Effective penetration of nanostructure across the cornea and other barriers is not well-documented in the literature. If there are strong evidence to support such penetration in the literature, they should be cited. Evidence of effective treatment (e.g., topical for dry eye disease) does not guarantee effective penetration of exosomes unless the mechanism is well defined. The statements in line 210 have the same issue.
2. The arrows and locations of the barriers in Fig. 1 are not correct. For example, the blood retinal barrier is pointing to the anterior segment, and the corneal barrier is pointing to the back of the eye.
3. The statement “the primary focus of the management of PDR is reducing the production of VEGF. . .” (line 524) is confusing. The main mechanism of current anti-VEGF injections in the treatment of neovascular disease is to bind and stop VEGF from binding to its receptor instead of reducing VEGF production.
4. The advantage of MSC-derived exosomes to overcome intravitreal injections is unclear. The statements “. . . injections also puts a burden on patient compliance . . . .” (line 626) require clarifications. Do the authors imply that the administration of MSC-derived exosomes does not require injection in the treatment of posterior eye diseases? The statements in line 729 have the same issue.
5. In the paper, it is unclear how MSC-derived exosomes can penetrate the barriers of the eye (e.g., line 770). The barriers such as blood-retinal barrier are used as a general term without any detailed descriptions and direct evidence (e.g., references are needed).
6. There are typos and grammatical errors that need to be corrected. Please proofread carefully.
Author Response
Thank you for taking the time to review our manuscript and for your positive feedback on the summary of MSC-derived exosomes in the treatment of anterior and posterior ocular diseases. We are glad to hear that you found our review timely and of importance.
We appreciate your point-by-point comments and have carefully considered each of them to improve the quality of the manuscript. We believe that your feedback will help us further refine and enhance the manuscript.
- The statement “nano-sized dimensions allow for their efficient delivery to target ocular structures. . . corneal barrier. . .” (line 46) requires justification. Effective penetration of nanostructure across the cornea and other barriers is not well-documented in the literature. If there are strong evidence to support such penetration in the literature, they should be cited. Evidence of effective treatment (e.g., topical for dry eye disease) does not guarantee effective penetration of exosomes unless the mechanism is well defined. The statements in line 210 have the same issue.
- Thank you for pointing out the need for justification regarding the statement on the efficient delivery of nano-sized dimensions to target ocular structures, including the corneal barrier (line 46). We acknowledge that the effective penetration of nanostructures across the cornea and other barriers is not well-documented in the literature, and we appreciate your suggestion to provide supporting evidence. In our revised manuscript, we have carefully reviewed the relevant literature and incorporated appropriate citations to substantiate our claims. We understand that evidence of effective treatment, such as topical application for dry eye disease, does not guarantee effective penetration of exosomes unless the mechanism is well-defined. We have ensured that our revised manuscript addresses this concern by providing a clearer explanation of the proposed mechanisms, based on the extrapolation of findings from their penetration through the blood-brain barrier (BBB). Additionally, we have revised the statements in line 210 to reflect the same level of clarification and evidence. Your valuable feedback helps us improve the quality and rigor of our work, and we are grateful for your guidance. Please do not hesitate to share any further concerns or suggestions.
- The arrows and locations of the barriers in Fig. 1 are not correct. For example, the blood retinal barrier is pointing to the anterior segment, and the corneal barrier is pointing to the back of the eye.
- We appreciate your attention to detail and your efforts in helping us improve the quality of our work. In response to your comment, we have carefully reviewed Figure 1 and have made the necessary changes to correct the arrows and locations of the barriers. We have exchanged the placement of the Corneal barrier and Blood-retinal barrier to follow the correct anatomical structure of the eye. We believe that this modification will improve the accuracy and clarity of the figure.
- The statement “the primary focus of the management of PDR is reducing the production of VEGF. . .” (line 524) is confusing. The main mechanism of current anti-VEGF injections in the treatment of neovascular disease is to bind and stop VEGF from binding to its receptor instead of reducing VEGF production.
- Thank you for your feedback on our manuscript and for bringing to our attention the issue with the statement in line 524. We appreciate your attention to detail and your efforts in helping us improve the quality of our work. In response to your comment, we have revised the statement in line 524 to clarify the primary focus of the management of PDR. The revised statement reads as follows: "The primary focus of the management of PDR is inhibiting the activity of VEGF in ischemic tissue through laser photocoagulation or intravitreal anti-VEGF injections, which work by binding to VEGF and preventing it from interacting with its receptor." We hope that this change clarifies the primary mechanism of current anti-VEGF injections in the treatment of neovascular disease and improves the overall accuracy of the manuscript.
- The advantage of MSC-derived exosomes to overcome intravitreal injections is unclear. The statements “. . . injections also puts a burden on patient compliance . . . .” (line 626) require clarifications. Do the authors imply that the administration of MSC-derived exosomes does not require injection in the treatment of posterior eye diseases? The statements in line 729 have the same issue.
- Despite the widespread use of anti-VEGF treatments, not all patients respond favora-bly, and there are potential vision-threatening complications such as endophthalmitis and retinal detachment. The reliance on frequent intravitreal injections also puts a bur-den on patient compliance. MSC-derived exosomes offer potential advantages in ad-dressing these issues: 1) they may significantly reduce the frequency of intravitreal in-jections due to better biocompatibility and longer duration of action resulting from pro-tection against degradation (as depicted in Figure 3), and 2) they have the potential to be delivered topically instead of intravitreally due to their capacity to penetrate through barriers and target specific tissues (as depicted in Figure 3). Therefore, opti-mizing therapies that target both inflammation and neovascularization with the use of MSC-derived exosomes could provide a more effective and less burdensome treatment solution [112,113].
- In the paper, it is unclear how MSC-derived exosomes can penetrate the barriers of the eye (e.g., line 770). The barriers such as blood-retinal barrier are used as a general term without any detailed descriptions and direct evidence (e.g., references are needed).
- Thank you for your insightful comment regarding the penetration of MSC-derived exosomes through the barriers of the eye. We appreciate your suggestion to provide a more detailed description and direct evidence to support our claims. In the revised manuscript, we have clarified how MSC-derived exosomes may potentially penetrate ocular barriers, such as the blood-retinal barrier, by drawing parallels with their demonstrated ability to cross the blood-brain barrier, as found in existing literature. We understand that our current discussion lacks specific references and detailed descriptions, and we have addressed this in the updated version of our paper.
- There are typos and grammatical errors that need to be corrected. Please proofread carefully.
- In response to your comment, we will carefully proofread the manuscript to identify and correct any typos and grammatical errors. We apologize for any errors that may have slipped through the editing process, and we will ensure that the final version of the manuscript is thoroughly checked for accuracy and clarity.
Reviewer 5 Report
In the review “A Vision for the Future: MSC-Derived Exosomes in Ocular 2 Therapy” the authors showed the potential of exosome-based therapies in different diseases, focusing on ocular pathology. The biological properties of MSCs-derived exosomes and their applications was widely described in the first part of the present work. The second part reported, with great accuracy, the main preclinical studies that have showed the potential of the exosomes in treating several ocular diseases. The Abstract is well structured and clear, including the article’s purpose, methods, and results.
In summary, the work’s purpose is intriguing and is explained in detail throughout the text.
Considerations for the author are about the style:
- The character of the references in the text are not the same, (e.g. line 327,334).
- The references' formats do not follow a uniform format throughout the text (line 537,544, 569, 696) .
Author Response
Thank you for taking the time to review our manuscript and for your positive feedback on our review article titled "A Vision for the Future: MSC-Derived Exosomes in Ocular Therapy." We are delighted to hear that you found the review to be well-structured and clear in terms of its purpose, methods, and results.
We appreciate your attention to detail and your efforts in helping us improve the quality of our work. In response to your comment, we will carefully review the manuscript to ensure that the character of the references in the text and the format of the references are consistent throughout the manuscript. We apologize for any inconsistencies that may have occurred, and we will ensure that the final version of the manuscript is presented in a clear and consistent format. We appreciate your helpful feedback and for your time in reviewing our work. Please let us know if you have any further suggestions or comments.